# Investigation of Whole and Glandular Saliva as a Biomarker for Alzheimer’s Disease Diagnosis

**DOI:** 10.3390/brainsci12050595

**Published:** 2022-05-03

**Authors:** Yangyang Cui, Hankun Zhang, Jia Zhu, Zhenhua Liao, Song Wang, Weiqiang Liu

**Affiliations:** 1Tsinghua Shenzhen International Graduate School, Tsinghua University, Shenzhen 518055, China; cuiyy20@mails.tsinghua.edu.cn (Y.C.); zhanghk20@mails.tsinghua.edu.cn (H.Z.); zhuj@tsinghua-sz.org (J.Z.); 2Department of Mechanical Engineering, Tsinghua University, Beijing 100084, China; 3Biomechanics and Biotechnology Lab, Research Institute of Tsinghua University in Shenzhen, Shenzhen 518057, China; liaozh@tsinghua-sz.org

**Keywords:** saliva, Alzheimer’s disease, diagnosis, biomarker, collection method

## Abstract

Salivary Aβ40, Aβ42, t-tau, and *p*-tau 181 are commonly employed in Alzheimer’s disease (AD) investigations. However, the collection method of these biomarkers can affect their levels. To assess the impact of saliva collection methods on biomarkers in this study, 15 healthy people were employed in the morning with six saliva collection methods. The chosen methods were then applied in 30 AD patients and 30 non-AD controls. The levels of salivary biomarkers were calculated by a specific enzyme-linked immunosorbent assay. The receiver operating characteristic was utilized to assess salivary biomarkers in AD patients. The results demonstrated that the highest levels of salivary Aβ40, Aβ42, t-tau, and *p*-tau were in different saliva collection methods. The correlations between different saliva biomarkers in the same collection method were different. Salivary Aβ40, Aβ42, t-tau, and *p*-tau had no significant association. Salivary Aβ42 was higher in AD than in non-AD controls. However, *p*-tau/t-tau and Aβ42/Aβ40 had some relevance. The area under the curve for four biomarkers combined in AD diagnosis was 92.11%. An alternate saliva collection method (e.g., USS in Aβ40, UPS in Aβ42, t-tau, SSS in *p*-tau 181) was demonstrated in this study. Moreover, combining numerous biomarkers improves AD diagnosis.

## 1. Introduction

Neurodegeneration is a multidimensional process involving multiple biochemical pathways and a complex interplay of a range of regulatory variables [1,2]. It is defined by the progressive and irreversible loss of neurons from certain brain and spinal cord regions, most notably the nuclei of the base within subcortical areas and the cerebral cortex, resulting in damage and dysfunction exhibited as cognitive and motor dysfunctions [3,4]. In general, the causative factors include oxidative stress and free radical formation; protein misfolding, oligomerization, and aggregation; mitochondrial dysfunction, axonal transport deficits, and abnormal neuron–glial interactions; calcium deregulation and phosphorylation impairment; neuroinflammation; DNA damage and aberrant RNA processing [5,6].

Numerous chronic and incurable age-related illnesses are caused by neurodegeneration [7]. The frequency of neurodegenerative illnesses is steadily increasing as a result of technological advancements and growth, posing a substantial threat to human health [8]. Additionally, neurodegeneration is associated with a variety of neurodegenerative, neurotraumatic, and neuropsychiatric disorders, each of which has a significantly different pathophysiology, including memory and cognitive impairments, muscle weakness and/or paralysis, abnormal voluntary movement control, seizures, confusion, and pain [9]. Such diseases range in severity from progressive degenerative disorders such as Alzheimer’s disease (AD), Parkinson’s disease, Huntington’s disease, amyotrophic lateral sclerosis, and multiple sclerosis, to acute traumatic injuries such as traumatic brain injury, stroke, or spinal cord injury [10,11].

AD is the most prevalent neurodegenerative condition, affecting about 45 million people globally and estimated to reach 60 million by 2030 as the elderly population grows [12]. It is the most common cause of dementia in late adulthood and is associated with a substantial socioeconomic cost, as well as increased morbidity and mortality. However, no effective treatment strategy for AD is currently available [13]. The primary risk factor for the majority of neurodegenerative disorders, including AD, is advancing age [14]. According to the WHO, dementia is a fast-expanding public health problem, affecting approximately 50 million people worldwide in 2019 [15], with AD accounting for 60% to 70% of cases. Due to the constraints of early AD diagnosis and frequent therapy delays, the therapeutic impact gradually deteriorates. Although lumbar puncture is the most often used procedure for diagnosing AD, its invasiveness can cause pain to the patient and is difficult for many individuals to accept [16,17]. AD is characterized by the progressive death of cholinergic neurons in the hippocampus and cortex, resulting in atrophy, neurotransmission anomalies and synaptic loss, and neurodegeneration [18]. The basic causes of AD entail the extracellular deposition of amyloid beta (Aβ) peptides and hyperphosphorylated tau protein aggregates. Intracellular production of neurofibrillary tangles that cause oxidative stress, persistent neuroinflammation, neuronal dysfunction, and neurodegeneration [19,20]. Jack et al. [21] discovered that the neuropathologic markers of AD may manifest ten to fifteen years before evident cognitive symptoms, which can be characterized as substantial memory concern, early mild cognitive impairment (MCI), or late MCI. It is possible to delay the onset of AD, alleviate patient suffering, and alleviate social burdens through early diagnosis and treatments [22]. The most recent guidelines stated unequivocally that biomarkers should be used as indicators for clinical diagnosis of AD, particularly in the early stages, and that this was almost the only method to do so [23,24]. As a result, it is critical to develop highly sensitive and specific biomarkers as well as more reliable and minimally intrusive methods for the early detection of AD.

Current diagnostic procedures, in addition to cognitive testing, rely on imaging techniques [25] and cerebrospinal fluid (CSF) measurements. Neuroimaging techniques use magnetic resonance imaging to assess hippocampal atrophy and positron emission tomography to assess cortical A deposition. Cerebrospinal fluid analysis, conversely, strives to offer quantitative assessments of Aβ and tau protein levels as biomarkers for AD [26,27]. Existing approaches are costly, rather invasive [28], and have low sensitivity and specificity, posing hazards of overdiagnosis or underdiagnosis, misattribution, or omission and neglect of symptoms [29]. Furthermore, due to a chronic paucity of AD diagnostic testing across all disease stages, individuals are frequently detected late, imposing a significant burden on health-care systems [30]. Saliva is a fluid that can be collected easily and noninvasively and has been recommended as a source of readily available biomarkers for the diagnosis and evaluation of a variety of pathological disorders, not only in the oral cavity, but also throughout the body [14,31]. Saliva glands produce saliva primarily in response to autonomic nervous system monitoring via cholinergic innervation of cranial nerves VII and IX [32]. These glandular secretions may reflect a variety of aspects of the nervous system’s physiology. Indeed, proteins from the central nervous system are released in an age-dependent way [33]. Additionally, proteins can enter saliva from the blood by passive diffusion, active transport, or microfiltration [34]. As a result of these findings, saliva may contain novel indicators of central nervous system impairment, making it a more convenient and accessible source for capturing AD-related biomarkers. Recent research indicates that saliva may be a source of noninvasive indicators for AD diagnosis. These indicators may be produced directly in salivary glands or may diffuse from blood. Additionally, salivary levels may indicate changes in CSF. Several publications have detected and quantified salivary Aβ40, Aβ42, t-tau, and *p*-tau levels in patients with AD and in controls [35,36].

Although saliva is a biomarker of human health, its application as a diagnostic fluid for AD has been disregarded due to a lack of standardized saliva collecting procedures. The majority of research use saliva as a diagnostic tool, employing a variety of collection methods and failing to clearly define the sampling protocols [37]. This complicates the task of comparing the findings of several investigations [38]. By and large, most research considers saliva to be a homogeneous bodily fluid. Saliva, however, is not a single fluid and cannot be regarded as one. Rather than that, it is a complicated mixture of the secretions of three major salivary glands (parotid, submandibular, and sublingual), each of which produces a distinct form of saliva, hundreds of minor salivary glands, and gingival crevicular fluids and debris. Additionally, it is unstable, and its composition is influenced by factors like as sampling technique, surroundings, dental cleanliness, and psychological state [39]. Due to the fact that numerous factors influence saliva output and composition, exact standards for saliva collection are important [40]. For example, the type of saliva samples, whether produced by whole saliva or specific glands, and whether the sample was collected following stimulation [41].

So, in this study, six saliva collection methods were employed in 15 healthy participants in the morning to choose the ideal saliva collection method. Then the chosen methods in different biomarkers were used on 30 AD patients and 30 non-AD controls. The levels of salivary Aβ42, Aβ40, t-tau, and *p*-tau were calculated by specific enzyme-linked immunosorbent assay (ELSA), and the receiver operating characteristic (ROC) method was used to evaluate the value of saliva biomarkers in clinically diagnosed AD.

## 2. Materials and Methods

### 2.1. Participants

In this study, first, 15 healthy participants (eight men and seven women), with a mean ± SD age of 67.5 ± 3.2 years old were included. Each participant had their saliva collected for 7 consecutive days with 6 different collection methods, and the mean value was utilized to determine the most optimal saliva collection method. Second, 30 patients diagnosed with AD were included. Selection criteria were [42] the diagnosis complied with the relevant standards formulated in the Chinese Mental Illness Classification Scheme and Diagnostic Criteria (CCMD-3) and the Diagnosis and Statistics Manual of Mental Illness. Another criterium was an age ≥ 60 years old. Exclusion criteria included severe organ dysfunction or having taken antipsychotic drugs a month before the experiment. Another 30 non-AD controls from the physical examination center during the same period were selected. Selection criteria included no mental illness, age ≥ 60 years old, no mental disorder or hereditary neurological disease among immediate family members, no history of head trauma and normal cognitive function.

### 2.2. Laboratory Tests

Thirty min before collection, study participants were told to refrain from smoking, cleaning their teeth, eating, or drinking. After that, the mouth was washed with water to remove any food remnants in the oral cavity [43]. Saliva was collected using a salivette (Sarstedt, 51.5134) (containing untreated swabs and swabs activated by citric acid), as well as six other collection methods [44].

With and without stimulation, samples of the parotid gland, mandibular/sublingual gland, and total saliva were taken from each participant. Unstimulated whole saliva (UWS), stimulated whole saliva (SWS), unstimulated sublingual/submandibular saliva (USS), stimulated sublingual/submandibular saliva (SSS), stimulated parotid saliva (SPS), and unstimulated parotid saliva (UPS)were collected. The collection time was between 9:00 and 9:30 a.m. In the same clinical room, all saliva samples were taken in the same order. To prevent delicate peptides from degrading, all samples were collected in pre-chilled polypropylene tubes on ice. The total amount of saliva obtained by all methods was 5 mL. Finally, it was transferred to the laboratory on a regular basis and centrifuged at −20 °C for subsequent use.

The six saliva collection methods were evaluated in the morning on fifteen healthy subjects to determine the optimal saliva collection method. The chosen procedures were then applied to 30 AD patients and 30 non-AD controls. Salivary Aβ40, Aβ42, t-tau, and *p*-tau levels were determined using a particular enzyme-linked immunosorbent test (ELISA, Beijing Furui Runkang Biotechnology Co., Ltd., Beijing, China). The saliva ELISA kit indicated a sensitivity of 0.537 ng/mL, a variation of 2.6% within the assay, and a variation of 6.6% between the assays.

### 2.3. Statistics

SPSS was used to perform statistical analysis. The data were expressed as relative numbers, and $\chi^{2}$ was used for comparison between groups. The measurement data was conformed to the normal distribution and expressed as mean ± standard deviation ($\bar{x}$ ± SD), and the *t*-test was used for comparison between groups. The Shapiro–Wilk test was used to test the normality of sample data. Non-normally distributed data were described in terms of minimum and maximum numbers, and normally distributed data were described in terms of ($\bar{x}$ ± SD). Hypothesis testing would have insufficient sensitivity when the sample size was small, which would cause the results to lose use value, and if the data deviated slightly from normality, the final test result would not have much impact, so box plots could also be combined to perform statistical analysis. *p* < 0.05 indicated that the difference was statistically significant.

## 3. Results

### 3.1. Sample Characteristics

Table 1 shows salivary Aβ40, Aβ42, t-tau, and *p*-tau levels of the studied groups, indicating that Aβ42 was the largest in UPS, Aβ40 was the largest in USS, t-tau was the largest in UPS, and *p*-tau was the largest in SSS. The maximum values of Aβ40, Aβ42, and t-tau were obtained in unstimulated saliva, however, the maximum value of *p*-tau was obtained in stimulated saliva.

Figure 1 is the summary of saliva data. It indicates the salivary Aβ40, Aβ42, t-tau, *p*-tau, Aβ42/Aβ40, and t-tau/*p*-tau concentrations with different saliva collection methods. It can be observed that the salivary Aβ42 and t-tau concentration of UPS were significantly higher than the other five methods. Concurrently, in salivary Aβ40, Aβ42, and t-tau, the stimulated saliva sample concentration was significantly lower than the unstimulated saliva sample concentration. However, the opposite conclusion was obtained in salivary *p*-tau. The maximum value of Aβ40/Aβ42 was obtained in UPS, however, the maximum value of *p*-tau/t-tau was obtained in SSS.

### 3.2. Correlation between Different Saliva Biomarkers

Figure 2 demonstrates the correlation between different saliva biomarkers, it can be observed that Aβ42 and Aβ42/Aβ40 in UWS, USS, UPS, SSS, SPS, and SWS were the most relevant. However, there is no significant correlation between Aβ40, Aβ42, t-tau, and *p*-tau.

The results demonstrated that the highest levels of salivary Aβ40, Aβ42, t-tau, and *p*-tau were in different saliva collection methods. The USS in Aβ40 had the highest levels. Similarly, UPS was in Aβ42, UPS in t-tau, and SSS in *p*-tau 181. The correlations between different saliva biomarkers in the same collection method were different. There was no significant correlation between Aβ40, Aβ42, t-tau, and *p*-tau. So, the chosen methods in different biomarkers were used on 30 AD patients and 30 non-AD controls.

### 3.3. Salivary Biomarker Levels across Diagnostic Groups

This study included 60 subjects (30 women and 30 men), with an average age of 72.3 ± 5.2 (60–89 years), composed of 30 patients (50%) and 30 controls (50%). There was no statistical difference in age or gender between the two groups. Table 2 described the saliva analytes of the study group and the results of univariate and multivariate logistic regression models regarding the effects of parameters on patients with dementia. In univariate analysis, individuals with higher Aβ42 levels were more likely to belong to the AD group. In the multivariate regression analysis, the levels of Aβ42/Aβ40 and *p*-tau/t-tau were similar, but the combination of *p*-tau and t-tau was better than a single factor being statistically significant.

Figure 3 shows the salivary Aβ40, Aβ42, t-tau, *p*-tau, *p*-tau/t-tau, and Aβ42/Aβ40 in AD patients and non-AD controls. Lines represent maximum and minimum range. It can be observed that based on the differential detection of Aβ42 between the AD group and the control group, where it had been found to be secreted in high concentrations in the saliva of individuals suffering from or at risk of developing AD, Aβ40 concentration had no significant change. However, the ratio of Aβ42/Aβ40 slightly increased, but not in an obviously relevant manner (*p* > 0.01). For *p*-tau and t-tau levels, the spearman rank analysis of saliva levels was not significant, and no significant change was observed, but *p*-tau/t-tau increased significantly. In general, the level of salivary Aβ42 of the AD group was significantly decreased, which was about 2.12 times higher.

### 3.4. Validation of Diagnostic Performance by ROC Curve

To verify the specificity, sensitivity, and accuracy of the saliva compositions for diagnosis of AD, the ROC curve analysis of the saliva compositions alone or in combination with diagnosis of AD was carried out. Figure 4 shows the ROC curve of saliva components in the diagnosis of AD. The results were as follows: the areas under the curve (AUC) of salivary Aβ40, Aβ42, t-tau, and *p*-tau alone to diagnose AD were 53.11% (*p* = 1.00) and 84.83% (*p* = 0.28), 50.50% (*p* = 0.23), and 58.38% (*p* = 0.31). The AUC of Aβ42/Aβ40 and *p*-tau/t-tau combined to diagnose AD were 64.77% and 63.44%, respectively. Among them, the AUC of 4 biomarkers combined in diagnosis of AD was the largest, equal to 92.11% (*p* = 0.000).

## 4. Discussion

The detection of AD from an early phase characterized by mild cognitive deterioration is very important, since only an early diagnosis can afford long-term symptom relief. Techniques capable of detecting the disorder in its early stages are thus urgently needed [45]. In this study, six saliva collection methods were employed in fifteen healthy participants in the morning to detect the ideal saliva collection method. Then the chosen methods in different biomarkers were used on 30 AD patients and 30 non-AD controls. ELISA was used to determine the levels of salivary Aβ40, Aβ42, t-tau, and *p*-tau of these participants. It was particularly important that we could detect the levels of salivary Aβ40, Aβ42, t-tau, and *p*-tau through a simple and reproducible method. This study also demonstrated the effectiveness of saliva as a diagnostic biological fluid depended on the standardization of collection methods to deliver the most accurate and meaningful results. Different saliva collection methods have a great influence on the concentration and correlation of Aβ40, Aβ42, t-tau, and *p*-tau. Therefore, the standardization of a saliva collection method is pivotal to minimizing the effect on the variations in saliva composition within and between individuals. The CSF biomarkers of AD reflect the key factors of the physiopathology of the disease and provide 90–95% sensitivity and specificity [46]. However, obtaining samples from CSF examination is a highly invasive and stressful process. For this reason, the search for less invasive methods to diagnose and monitor AD is gaining attention. Compared with CSF, blood measurements are advantageous for AD biomarker screening because blood collection is easier and less invasive. However, drawing blood can make some people squeamish. Ranging from a slight discomfort to creating a panicked and fearful state, some people would rather not have blood work done. Blood tests are not always definitive. Sometimes instead of providing a solution and answer, they instead raise more questions. Additionally, waiting for the results of blood test can generate a great amount of anxiety [47]. Conversely, saliva sampling is noninvasive, safe, cheap, and easily performed compared to blood [48]. This fact makes it a biological fluid for the research and monitoring of biomarkers for various diseases. However, there are few studies of the behavior of saliva biomarkers and their utility in AD.

Saliva is produced by three major salivary glands (parotid, submandibular, and sublingual) and numerous minor salivary glands. Saliva contains a myriad of salivary proteins which could serve as biological markers for diagnosing and tracking the progression of various health conditions, as well as monitoring the effectiveness of medication [49,50]. To date, most of the saliva collection devices that are commercially available allow a person to collect resting/unstimulated saliva and/or stimulated saliva either via mechanical stimulation or acid stimulation [51]. When a person is in a resting state, saliva production is largely produced by the submandibular gland, while only 20% and 8% are produced by parotid and sublingual glands, respectively [52]. In contrast, when saliva production is stimulated through acid stimulation, most of the saliva produced is primarily derived from the parotid gland [50]. Most importantly, the compositions of both stimulated and unstimulated saliva may be altered by genetic predisposition factors and physiological, pathological, and environmental factors [53]. All these factors may hinder the correct derivation of results for best care outcomes. Therefore, this study determined the possible optimal collection method for each marker through the collection and detection of saliva markers from 15 elderly healthy individuals for 7 consecutive days.

The relative contribution of different glands to the whole sample of saliva varies with the collection method, the degree of stimulation, age, and even the time of day [34]. The variable nature of saliva secretion suggests that different methods may have to be used when studying its components or their possible role as indicators of specific physiological conditions. There is a large amount of literature on the diagnostic possibilities of saliva, but there is still no standardized method for collecting saliva samples. In different studies, different sampling methods are often used, and many studies do not or rarely describe patient preparation or sampling procedures [54]. In addition, without proper clinical examination, the characteristics of participants are usually insufficient. Most saliva Aβ40, Aβ42, t-tau, and *p*-tau research papers focus on studying the whole saliva [55] because it can be easily obtained by spitting it into a test tube or letting it drip from the mouth. Few people pay attention to ductal saliva obtained from different salivary glands. Moreover, comparing the Aβ40, Aβ42, t-tau, and *p*-tau expression of whole saliva and glandular saliva was focused on a cohort with careful characterization and clinical examination. The results indicate that different collection methods provide significant differences in the snapshots of salivary Aβ40, Aβ42, t-tau, and *p*-tau. In this study, the results indicate that different saliva collection methods provide significant differences in the snapshots of saliva biomarkers for AD. Based on the comparison of unstimulated and stimulated saliva collection methods, it can be concluded that, based on the simplicity and low variability of the collection method, different biomarkers may prefer different collection methods. Therefore, standardization of saliva collection method is pivotal to minimising the effect on the variations in saliva composition within and between individuals. The alternative saliva collection methods (e.g., USS in Aβ40, UPS in Aβ42, UPS in t-tau, SSS in *p*-tau 181) would be an ideal way to collect saliva in a clinical, challenging environment. It was particularly important that we could detect the levels of salivary Aβ40, Aβ42, t-tau, and *p*-tau through a simple and reproducible method. In the present study, we determined the levels of salivary Aβ40, Aβ42, t-tau, and *p*-tau using six saliva collection methods including UWS, SWS, USS, SSS, SPS, and UPS. Different saliva collection methods have a great influence on the concentration and correlation of Aβ40, Aβ42, t-tau, and *p*-tau.

We found, compared with the control group, statistically significant increased levels of salivary Aβ42 in AD patients. In addition, according to the ROC analysis performed for the predictive value of Aβ42 levels, the AUC was 0.848. In addition, salivary Aβ40 expression was unchanged within all the studied samples. We also analyzed the ratio of salivary Aβ42/Aβ40 and we found that, compared with the control group, AD patients were higher, but not enough to be statistically significant (*p* = 0.2). In general, compared with the control group, the level of salivary Aβ42 of early AD patients was significantly increased, which indicated that measuring salivary Aβ42 can be used as a biomarker to identify and confirm early AD diagnosis.

The miscleavage of amyloid precursor protein (APP), the pathological accumulation of which underpins AD, is the genesis of Aβ plaques. APP is cleaved into soluble APP by secretase under physiological conditions, which is then cleaved into p3 peptide and APP intracellular domain by secretase [56]. APP has been established in studies to have a key function in brain homeostasis as well as neuronal growth and maturation during brain development [57]. In AD, APP is cleaved by beta-secretase and gamma-secretase rather than alpha-secretase. This enzymatic breakage cascade produces amyloid Aβ40 and Aβ42 peptides, which collect in the extracellular space and form plaques, causing neurotoxicity and activating reactive inflammatory processes, which eventually lead to neuronal damage [58]. This amyloidogenic pathway is a well-known source of diagnostic biomarkers for AD. Aβ deposits detectable by PET scans, and Aβ levels in cerebrospinal fluid, as well as other body fluids, are utilized as a diagnostic technique for AD The levels of Aβ40 and Aβ42 are the most reliable techniques for AD diagnosis among the numerous A subtypes. Aβ42 specifically accumulates form plaques in the brain, and its concentration is in cerebrospinal fluid falls, which is a sign of AD. Although Aβ40 was the most prevalent isoform, its levels in Alzheimer’s patients did not decrease appreciably. As a result, our study concentrated on Aβ42 and Aβ40.

Therefore, it is particularly important to identify stable and reproducible salivary Aβ42 expression. Because it can be used as a potential indicator of AD neuropathology, it can be measured with the least stress on the subject. The mechanism of salivary Aβ42 accumulation is unclear. The localization may be due to the release of the peptide from the saliva gland due to the secretion of secretase from the saliva gland epithelial cells during the processing of amyloid precursor protein (APP) [59]. The results of this study are consistent with previous studies using CSF and plasma samples [60,61]. The Aβ42 content of AD patients is approximately 2.12 times that of the control group. Studies have reported that Aβ42 is reduced in patients’ CSF [62], and it can be understood that Aβ in the AD group is due to the damage of the blood brain barrier, which leads to the loss of neurons that produce APP, and the accumulation of Aβ in the brain and the reduction of Aβ clearance. Since the latter does not exist in the saliva glands, the Aβ produced by the APP glands may accumulate in the saliva [63].

There are various plausible causes for the greater agreement between amyloid PET and salivary Aβ42 when the Aβ42/Aβ40 ratio is used instead of the Aβ42/Aβ40 ratio. It is possible that non-AD subcortical lesions cause high salivary Aβ42 levels. Several studies [64,65] have discovered elevated amounts of salivary Aβ42 in multiple system atrophy and multiple sclerosis. Another possibility is variances in overall Aβ production and clearance leading to interindividual variability in total salivary Aβ levels. This is reinforced by recent observations indicating Aβ42 associated with Aβ40 in saliva, even in healthy controls. As a result, when salivary Aβ42 was utilized to detect AD brain pathology, the Aβ40 ratio used may have an effect on interindividual variability in total Aβ levels [66].

We found that compared with the control group, no difference in salivary t-tau and *p*-tau (using T181) concentration was found between AD patients and non-AD controls, which was consistent with the results obtained by Albahri et al. [67]. We analyzed the ratio of salivary *p*-tau/t-tau and found that there were significant differences between AD patients and non-AD controls, which was consistent with the results obtained by Pekeles et al. [68]. Several explanations for changes in the *p*-tau/t-tau ratio levels in AD and controls are worth considering. A small number of people who are clinically diagnosed with possible AD may not have tau pathology themselves [69]. Another explanation may be that even for those AD subjects who demonstrated elevated tau levels in the brain, some of these people still failed to express tau peripherally in the saliva gland tissue. Alternatively, it is possible that abnormal tau is present in most saliva tissues, but only a subgroup of AD individuals secreted abnormal tau in saliva.

Tau proteins are members of the microtubule-associated protein family and, by binding to tubulin, contribute to microtubule stability and flexibility [70]. Tau phosphorylation normally promotes its disintegration from microtubules, as well as its instability and removal [71]. Mutations in the tau protein sequence have been suggested to change its phosphorylation site, resulting in tau hyperphosphorylation [72]. *p*-tau can combine and form neurofibrillary tangles within cells. Lau et al. [73] evaluated t-tau and *p*-tau. The findings revealed that there were no quantitative differences between t-tau and *p*-tau patients in AD patients and healthy participants. However, in AD patients, there was a modest increase in *p*-tau levels (*p* < 0.05), which is consistent with the findings of this study. Min Shi et al. [74] investigated *p*-tau as a salivary biomarker for AD, and analysis of tau protein species, namely *p*-tau and t-tau, revealed that the *p*-tau/t-tau ratio of each individual was higher when compared to the healthy control group, which is consistent with the findings of this study. This demonstrates that an increase in phosphorylated tau concentration relative to total tau concentration could be employed as a possible biomarker to detect AD [75].

Detection techniques based on novel biomarkers other than Aβ and tau protein may be a potential approach for the early detection of AD [76]. However, because no single biomarker can effectively diagnose AD, a combination of biomarkers can considerably enhance diagnostic accuracy [77,78]. Although Aβ plaque deposition occurs years or even decades before symptom onset and can be utilized to make an early diagnosis, tau biomarkers alter with disease development and are closely linked to local degeneration and cognitive loss [79,80]. Combining disease-specific and nonspecific biomarkers is the most effective technique for building biomarker-based diagnostic tools. In this context, the reduction in Aβ42, along with alterations in the Aβ42/Aβ40 ratio, t-tau, and *p*-tau levels, is commonly referred to as an Alzheimer’s signature because it allows for the detection of AD at an early stage [81]. Furthermore, their combined usage for AD diagnosis has a sensitivity and specificity of about 85–95% [80]. This is consistent with the study’s result that when four markers are used together to predict AD, the accuracy is the highest (92.11%). As a result, combining biomarkers can enhance prediction accuracy.

However, our study has limitations. Firstly, when studying several biomarkers, they have not been compared with gold standards such as CSF concentration and imaging. Although this study was conducted in patients who have already been diagnosed, the saliva biomarkers that reflect the current situation cannot be compared with image data obtained in some cases a few years ago. Therefore, the patient’s saliva biomarkers should be diagnosed when imaging or CSF is obtained for AD comparison to evaluate their salivary biomarkers. In addition, clinical studies have described the excellent diagnostic performance of saliva biomarkers. Although this study provides the best performance for early diagnosis of patients with cognitive problems and suspected AD, due to the small amount of data, further development is still needed, including validity testing, retesting, and multifactor testing. Finally, there are still many problems in this study that need to be resolved and further explored. For example, the submandibular glands and sublingual glands are closely located, so it is difficult to separate saliva from these glands with certainty, which was why saliva was collected from both glands in the current study. How to distinguish sublingual saliva from submandibular saliva is also a direction that needs further research. Finally, various risk factors (e.g., the role of ventromedial prefrontal cortex in the processing of safety-threat information and their relative value [82]) should also be included in the diagnosis of AD in saliva to make the diagnosis more accurate. Providing a deeper understanding of human learning neural networks, particularly on human processing of safety crucial role, is also the focus of future research. This might also contribute to the advancement of alternative, more precise and individualized treatments for psychiatric disorders [83].

## 5. Conclusions

In this study, the results indicated that different saliva collection methods provided significant differences in the snapshots of saliva biomarkers for AD. Based on the comparison of unstimulated and stimulated saliva collection methods, it can be demonstrated that, based on the simplicity and low variability of the collection method, different biomarkers may prefer different collection methods. Therefore, standardization of a saliva collection method is pivotal to minimising the effect on the variations in saliva composition within and between individuals. The alternative saliva collection methods (e.g., USS in Aβ40, UPS in Aβ42, UPS in t-tau, SSS in *p*-tau 181) would be an ideal way to collect saliva in a clinical, challenging environment. This study also demonstrated that salivary biomarkers can be quantified and used to diagnose AD. We found significantly higher Aβ42 levels in the AD group compared to the control group. Aβ40, t-tau, and *p*-tau did not change significantly, however, *p*-tau/t-tau and Aβ42/Aβ40 had a certain relevance. Among them, the AUC of 4 biomarkers combined in the diagnosis of AD was large, equaling 92.11%. Generally, these results emphasize the importance of consistency when collecting saliva samples, which should be more important than the collection method itself. Our findings from this study pave the way towards making saliva diagnostics a reality for AD. With further research and standardization of collection and quantification methods with larger sample groups, various risk factors (e.g., the role of ventromedial prefrontal cortex in the processing of safety-threat information and their relative value [82]) should also be included in the diagnosis of AD in saliva to make the diagnosis more accurate. saliva biomarkers may become the gold standard for early diagnosis and screening of AD. However, our study has limitations. In future, this study will be to further validate and compare the salivary biomarkers to existing and currently used biomarkers of AD progression that will lend more credence to these studies.

## Figures and Tables

**Figure 1 brainsci-12-00595-f001:**
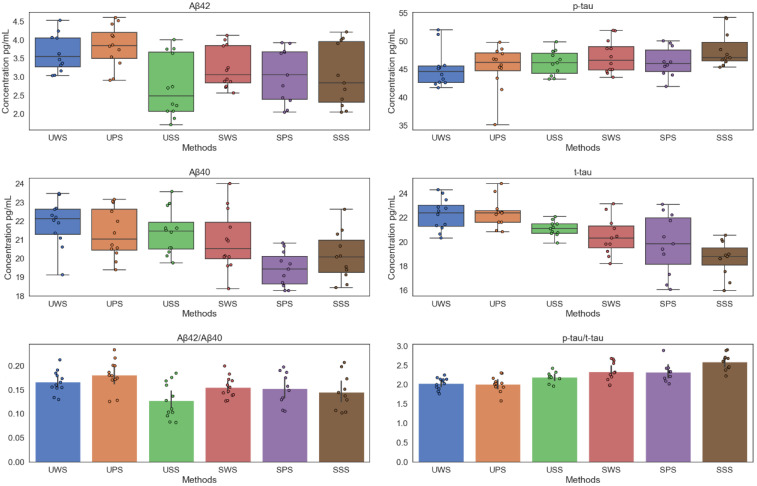
Salivary Aβ40, Aβ42, t-tau, *p*-tau, Aβ40/Aβ42, and t-tau/*p*-tau concentrations at different saliva collection methods.

**Figure 2 brainsci-12-00595-f002:**
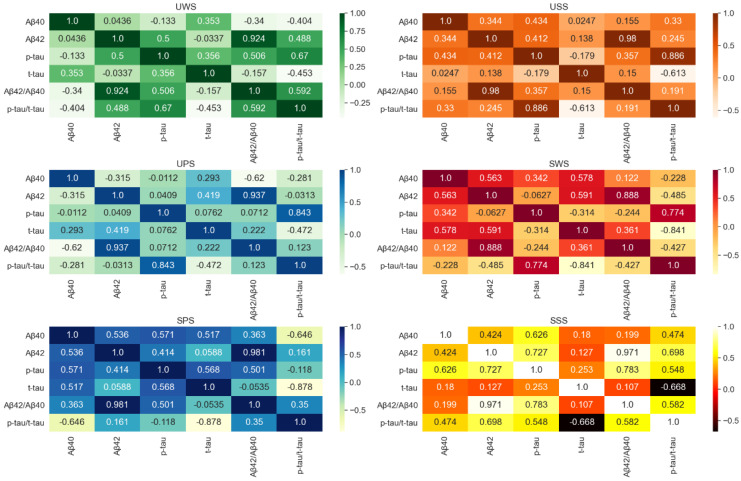
The correlation between different saliva biomarkers.

**Figure 3 brainsci-12-00595-f003:**
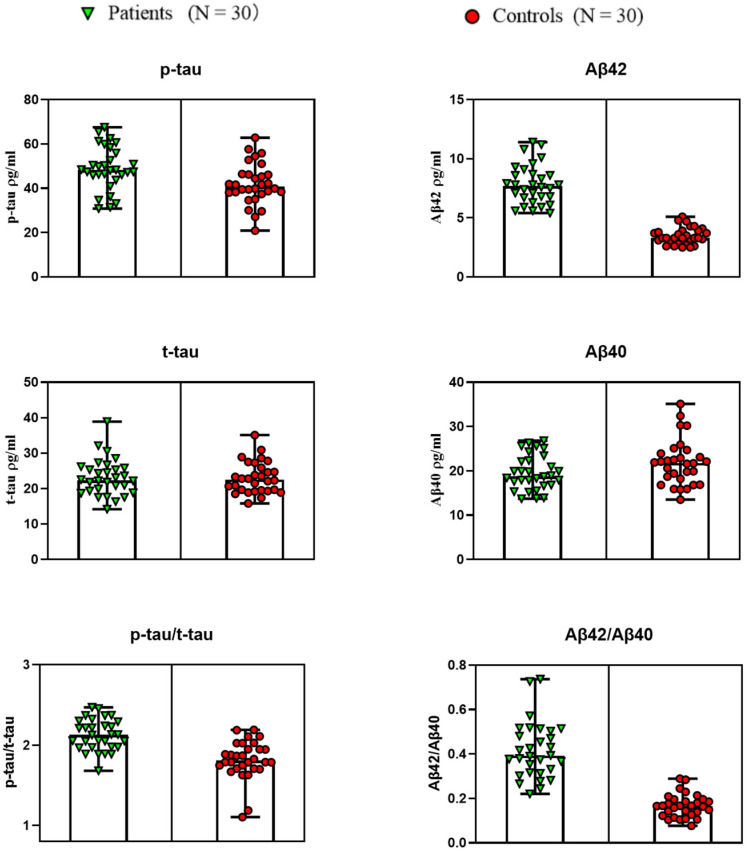
Salivary Aβ40, Aβ42, t-tau, *p*-tau, *p*-tau/t-tau, and Aβ42/Aβ40 in AD patients and non-AD controls. Lines represent the maximum and minimum range.

**Figure 4 brainsci-12-00595-f004:**
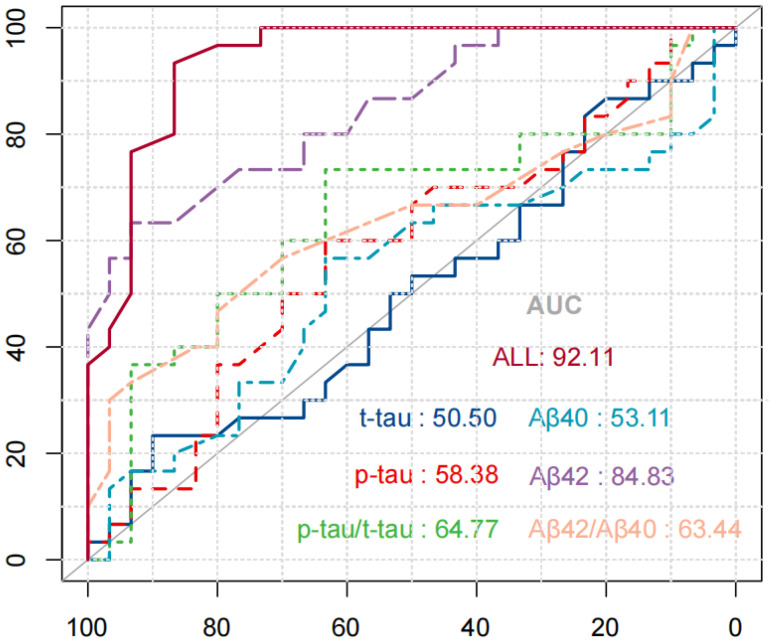
ROC curve of saliva components in the diagnosis of AD.

**Table 1 brainsci-12-00595-t001:** Salivary Aβ40, Aβ42, t-tau, and *p*-tau levels of the studied groups.

Methods	UPS	SPS	USS	SSS	UWS	SWS
Aβ42 pg/mL	3.83 ± 0.54 *	3.05 ± 0.71	2.73 ± 0.80	3.04 ± 0.81	3.62 ± 0.47	3.25 ± 0.53
Aβ40 pg/mL	21.36 ± 1.28	19.43 ± 0.88	21.41 ± 1.15 *	20.13 ± 1.23	21.91 ± 1.17	20.94 ± 1.55
*p*-tau pg/mL	45.36 ± 3.78	46.31 ± 2.50	46.18 ± 2.11	48.51 ± 3.026 *	45.13 ± 3.17	47.25 ± 2.82
t-tau pg/mL	22.39 ± 1.16 *	19.83 ± 2.36	21.09 ± 0.59	18.62 ± 1.37	22.27 ± 1.24	20.46 ± 1.47

SD is standard deviation; * is the maximum value of each biomarker.

**Table 2 brainsci-12-00595-t002:** Saliva biomarkers in AD.

Measure	Patients		Controls	
	95% CI	*p*	95% CI	*p*
Aβ42	0.91	0.215	0.93	0.350
Aβ40	0.85	0.287	0.87	0.507
Aβ42/Aβ40	0.87	0.347	0.89	0.460
*p*-tau	0.79	0.877	0.81	0.398
t-tau	0.85	0.311	0.87	0.786
*p*-tau/t-tau	0.84	0.435	0.87	0.501

## Data Availability

The study did not report any data.

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
