# Peer review of "Investigation of Whole and Glandular Saliva as a Biomarker for Alzheimer’s Disease Diagnosis"

_brainsci, 2022, doi:10.3390/brainsci12050595_

Round 1

Reviewer 1 Report

Comparing saliva collection methods should be a paragraph in the Materials and Methods or Supplementary section of an actual research paper, not actually comprise a whole paper.

Apart from this fact, the data they do present is not persuasive, as they don't even compare it to biomarker levels from another biological sample, e.g. plasma, to see if the results are even accurate in any way. 

Author Response

Thank you for your careful review and valuable suggestions. Salivary Aβ40, Aβ42, t-tau, and p-tau 181 were widely used in Alzheimer’s disease (AD) studies. However, saliva collection methods can influence the levels of these salivary biomarkers. This study aims to evaluate the effect of saliva collection methods on salivary Aβ40, Aβ42, t-tau, and p-tau 181 and identify the ideal saliva collection method. Because the effect of different collection methods on the content of saliva substances is very large and affects the use of saliva for disease diagnosis results, this study focuses on the effect of collection methods on the content of saliva substances, so it is not the material and methods of the actual research paper or in the supplementary section. At the same time, our main goal is to compare the effect of different collection methods on the content of substances in saliva, as well as to emphasize the importance of collection methods, so we did not compare with other substances, but this will not affect the overall results of the study. In this study, six saliva collection methods were employed in 15 healthy participants in the morning to choose the ideal saliva collection method. Then the chosen methods in different biomarkers were used of 30 AD patients and 30 non-AD controls. The levels of salivary Aβ42, Aβ40, t-tau and p-tau 181 were calculated by specific enzyme-linked immunosorbent assay (ELISA). The receiver operating characteristic was used to evaluate the value of saliva biomarkers in clinically diagnosed AD. The results showed that the highest levels of salivary Aβ40, Aβ42, t-tau, and p-tau were in different saliva collection methods. The unstimulated sublingual/submandibular saliva (USS) in Aβ40 had highest levels. Similarly, unstimulated parotid saliva (UPS) in Aβ42, unstimulated parotid saliva (UPS) in t-tau, and stimulated sublingual/submandibular saliva (SSS) in p-tau 181. The correlations between different saliva biomarkers in the same collection method were different. There was no significant correlation between Aβ40, Aβ42, t-tau and p-tau. Salivary Aβ42 was significantly higher in AD compared to non-AD controls. Meanwhile, Aβ40, t-tau, and p-tau did not change significantly, however, p-tau/t-tau and Aβ42/Aβ40 had a certain relevance. Among them, the area under the curve of 4 biomarkers combined in diagnosis of AD was the largest, equal to 92.11%. This study demonstrated that alternative saliva collection methods (e. g. USS in Aβ40, UPS in Aβ42, UPS in t-tau, SSS in p-tau 181) would be an ideal way to collect saliva in a clinical. Besides, combination of multiple biomarkers makes Alzheimer's diagnosis more precise. So, our conclusion is convincing. Finally, we have optimized the article again this time, the grammar errors, sentence structure and sentence fragments, etc. had been revised in revised manuscript. We would like to re-submit the revised manuscript for your consideration. We hope that the revision is acceptable for publication in your journal.

Reviewer 2 Report

Beautiful work, and well written. 

Congratulations

Author Response

Point 1: Beautiful work, and well written.

Congratulations.

Response 1: Thank you for your careful review and valuable suggestions. We would like to express our sincere thanks again to you for the constructive and positive comments on our manuscript entitled “Validation of whole and glandular saliva as a biomarker for Alzheimer’s disease diagnosis” (Manuscript ID: brainsci-1676756). We have optimized the article again this time, the grammar errors, sentence structure and sentence fragments, etc. had been revised in revised manuscript.

Reviewer 3 Report

Cuia and colleagues in the present study entitled ‘Validation of whole and glandular saliva as a biomarker for Alzheimer’s disease diagnosis’, verified the feasibility of saliva as a diagnosis in early diagnosis of AD. For this purpose, to evaluate the effect of saliva collection methods on salivary Aβ40, 11 Aβ42, t-tau, and p-tau 181 and identify the ideal saliva collection method, the authors tested six saliva collection methods in healthy participants, and then the chosen best methods were used in 30 AD patients and in 30 non-AD controls; finally, levels of salivary Aβ42, Aβ40, t-tau and p-tau 181 were calculated by specific enzyme-linked immunosorbent assay (ELISA). Results showed that alternative saliva collection methods (e. g. USS in Aβ40, UPS in Aβ42, UPS in t-tau, SSS in p-tau 181 would be an ideal way to collect saliva in a clinical. The authors concluded by stating that the combination of multiple biomarkers makes Alzheimer's diagnosis more precise.

The main strength of this manuscript is that it addresses an interesting and timely question, providing a captivating interpretation and describing how saliva biomarkers can be quantified and used to diagnose AD. In general, I think the idea of this research article is really interesting and the authors’ fascinating observations on this timely topic may be of interest to the readers of Brain Sciences. However, some comments, as well as some crucial evidence that should be included to support the authors’ argumentation, needed to be addressed to improve the quality of the manuscript, its adequacy, and its readability prior to the publication in the present form. My overall judgment is to publish this research article after the authors have carefully considered my suggestions below, in particular reshaping parts of the Introduction and Discussion sections by adding more evidence.

Please consider the following comments:

  • Abstract: According to the Journal’s guidelines, the abstract should be a total of about 200 words maximum and should be introduced as a single paragraph, following the style of structured abstracts, but without headings. Please correct the actual one.
  • In general, I recommend authors to use more evidence to back their claims, especially in the Introduction of the review, which I believe is currently lacking. Thus, I recommend the authors to attempt to deepen the subject of their manuscript, as the bibliography is too concise: nonetheless, in my opinion, less than 50 articles for a literature review are really insufficient. Indeed, currently, authors cite only 39 papers, and they are too low. Therefore, I suggest the authors to focus their efforts on researching more relevant literature: I believe that adding more studies and reviews will help them to provide better and more accurate background to this study.
  • Introduction: The ‘Introduction’ section is well-written and nicely presented, with a good balance of information about molecular pathophysiology and neurological changes associated with Alzheimer’s disease. Nevertheless, I believe that more information about structural and functional changes associated with neurodegeneration in AD will provide a better and more accurate background. Thus, I suggest the authors to make such an effort to provide a brief overview of the pertinent published on neurobiological signs of AD, because as it stands, this information is not highlighted in the text. In this regard, I believe that the statement ‘…neuropathologic markers of AD may manifest ten to fifteen years before evident cognitive symptoms, which can be characterized as substantial memory concern, early mild cognitive impairment (MCI), or late MCI needs some necessary citations. In particular, according with this sentence, I would recommend citing a recent review that examined pathophysiological basis and biomarkers of AD pathology and investigated molecular signs of neuroinflammation in neurodegenerative diseases, in particular Alzheimer’s disease (https://doi.org/10.3390/ijms21072431). Importantly, I also recommend a relevant study in which author investigated age-related impairments in the ability to process contextual information and in the regulation of responses to threat, addressing that structural and physiological alterations in the prefrontal cortex and medial temporal lobe determine cognitive changes in advanced aging, that can eventually cause patterns of cognitive dysfunctions observed in patients with AD/MCI (https://doi.org/10.1038/s41598-018-31000-9). I firmly believe that these improvements will help to provide a more coherent and defined background.
  • Participants: I suggest the authors to reshape this paragraph because, as it is, it may be difficult for the reader to fully comprehend that the first 15 controls were used to select the best salive collection method.
  • Laboratory test: I suggest the authors to rewrite this paragraph, because as it stands, this section is written with an informal style, and appears to be far too much impersonal and dispersive.
  • Results: In my opinion, this section is well organized, but it illustrates findings in an excessively broad way, without really providing full statistical details, to ensure in-depth understanding and replicability of the findings. Indeed, I believe that it is necessary for the authors to present their findings with a precise description not just in the in descriptive tables, but also in the main text.
  • Discussion: In this final section, authors described the results and their argumentation and captured the state of the art well; however, I would have liked to see some views on a way forward. I believe that the authors should make an effort, trying to explain the theoretical implication as well as the translational application of this research article, to adequately convey what they believe is the take-home message of their study. Discussion of theoretical and methodological avenues in need of refinement is necessary, as well as suggestions of a path forward in the understanding of supportive interventions and treatment for people with dementia. In particular, since the focus of this research was on Aβ, t-tau and p-tau levels in AD, I would suggest adding some studies that might discuss amyloid-β (Aβ) pathology in AD, highlighting the combined effect of forms of Aβ and tau protein to drive healthy neurons into the diseased state. Aβ peptide and tau protein consistently accumulate in the frontal and/or parietal lobes, and cause alterations of frontal lobe that impact memory and error-driven learning in individuals who have a high risk of dementia: evidence from a recent theoretical review (https://doi.org/10.1038/s41380-021-01326-4) that focused on neurobiology of fear conditioning, analyzed the role of ventromedial prefrontal cortex (vmPFC) in the processing of safety-threat information and their relative value, and how this region is fundamental for the evaluation and representation of stimulus-outcome’s value needed to produce sustained physiological responses. Also, I believe that a recent yet relevant perspective manuscript (https://doi.org/10.17219/acem/146756) might be of interest: here the focus was on providing a deeper understanding of human learning neural networks, particularly on human PFC crucial role, that might also contribute to the advancement of alternative, more precise and individualized treatments for psychiatric disorders. Secondary, authors also might to consider some studies that have focused on this topic (https://doi.org/10.3390/biomedicines10010076; https://doi.org/10.3390/biomedicines9050517). I believe that adding information from these studies may improve the interpretation of the results of the present article and their argumentation by highlighting how cognitive alterations caused by frontal dysfunction are fundamental as neurodegenerative biomarkers of AD.
  • In my opinion, I think the ‘Conclusions’ paragraph would benefit from some thoughtful as well as in-depth considerations by the authors, because as it stands, it is very descriptive but not enough theoretical as a discussion should be. Authors should make an effort, trying to explain the theoretical implication as well as the translational application of their research.
  • In according to the previous comment, I would ask the authors to include a ‘Limitations and future directions’ section before the end of the manuscript, in which authors can describe in detail and report all the technical issues brought to the surface.
  • Figures: I suggest to modify the Figures 2 and 4 for clarity because, as it stands, the readers may have difficulty comprehending it. In my opinion, authors should provide higher-resolution image of brain areas showed in the right panel, to allow a better detection of source regions of peak intensity. Also, please change the scale of the vertical axis and use the same minimum/maximum scale value in all the graphs.
  • References: Authors should consider revising the bibliography, as there are several incorrect citations. Indeed, according to the Journal’s guidelines, they should provide the abbreviated journal name in italics, the year of publication in bold, the volume number in italics for all the references.

Overall, the manuscript contains 5 figures, 2 tables and 39 references. In my opinion, the number of references it is dramatically low for an original research article, and this prevents the possibility of publishing it in this form. References should be more than 50 for original research articles. However, the manuscript might carry important value describing how saliva biomarkers can be quantified and used to diagnose AD.

I hope that, after these careful revisions, the manuscript can meet the Journal’s high standards for publication. I am available for a new round of revision of this article.

Best regards,

Reviewer

Author Response

Point 1: Cuia and colleagues in the present study entitled ‘Validation of whole and glandular saliva as a biomarker for Alzheimer’s disease diagnosis’, verified the feasibility of saliva as a diagnosis in early diagnosis of AD. For this purpose, to evaluate the effect of saliva collection methods on salivary Aβ40, 11 Aβ42, t-tau, and p-tau 181 and identify the ideal saliva collection method, the authors tested six saliva collection methods in healthy participants, and then the chosen best methods were used in 30 AD patients and in 30 non-AD controls; finally, levels of salivary Aβ42, Aβ40, t-tau and p-tau 181 were calculated by specific enzyme-linked immunosorbent assay (ELISA). Results showed that alternative saliva collection methods (e. g. USS in Aβ40, UPS in Aβ42, UPS in t-tau, SSS in p-tau 181 would be an ideal way to collect saliva in a clinical. The authors concluded by stating that the combination of multiple biomarkers makes Alzheimer's diagnosis more precise.

The main strength of this manuscript is that it addresses an interesting and timely question, providing a captivating interpretation and describing how saliva biomarkers can be quantified and used to diagnose AD. In general, I think the idea of this research article is really interesting and the authors’ fascinating observations on this timely topic may be of interest to the readers of Brain Sciences. However, some comments, as well as some crucial evidence that should be included to support the authors’ argumentation, needed to be addressed to improve the quality of the manuscript, its adequacy, and its readability prior to the publication in the present form. My overall judgment is to publish this research article after the authors have carefully considered my suggestions below, in particular reshaping parts of the Introduction and Discussion sections by adding more evidence.

Please consider the following comments:

Response 1: Thank you for your careful review and valuable suggestions. We appreciate editor and reviewers very much again for your positive and constructive comments and suggestions on our manuscript entitled “Validation of whole and glandular saliva as a biomarker for Alzheimer’s disease diagnosis” (Manuscript ID: brainsci-1676756). We have studied the comments of both the editor and reviewers carefully and tried our best to revise the paper accordingly. Point-to-point replies are included as below.

Point 2: Abstract: According to the Journal’s guidelines, the abstract should be a total of about 200 words maximum and should be introduced as a single paragraph, following the style of structured abstracts, but without headings. Please correct the actual one.

Response 2: Thank you for your careful review and valuable suggestions. According to the Journal’s guidelines, the abstract was corrected to be a total of about 195 words and it was introduced as a single paragraph, following the style of structured abstracts, and without headings, as followed. (Lines 14-30).

Salivary Aβ40, Aβ42, t-tau, and p-tau 181 were commonly employed in Alzheimer’s disease (AD) investigations. However, the collection method of these biomarkers can affect their levels. To assess the impact of saliva collection methods on biomarkers.  In this study, 15 healthy people were employed in the morning with six saliva collection methods. The chosen methods were then applied in 30 AD patients and 30 non-AD controls. The levels of salivary biomarkers were calculated by a specific enzyme-linked immunosorbent assay. The receiver operating characteristic was utilized to assess salivary biomarkers in AD patients. The results showed that the highest levels of salivary Aβ40, Aβ42, t-tau, and p-tau were in different saliva collection methods. The correlations between different saliva biomarkers in the same collection method were different. Salivary Aβ40, Aβ42, t-tau, and p-tau had no significant association. Salivary Aβ42 was higher in AD than in non-AD controls. However, p-tau/t-tau and Aβ42/Aβ40 had some relevance. The area under the curve for four biomarkers combined in AD diagnosis was 92.11%. An alternate saliva collection method (e. g. USS in A40, UPS in A42, t-tau, SSS in p-tau 181) was demonstrated in this work. Moreover, combining numerous biomarkers improves AD diagnosis.

Point 3: In general, I recommend authors to use more evidence to back their claims, especially in the Introduction of the review, which I believe is currently lacking. Thus, I recommend the authors to attempt to deepen the subject of their manuscript, as the bibliography is too concise: nonetheless, in my opinion, less than 50 articles for a literature review are really insufficient. Indeed, currently, authors cite only 39 papers, and they are too low. Therefore, I suggest the authors to focus their efforts on researching more relevant literature: I believe that adding more studies and reviews will help them to provide better and more accurate background to this study.

Response 3: Thank you for your careful review and valuable suggestions. In order to make this study clearer to the reader, and to provide a better and more accurate background to this study, we added more evidence to back our claims, especially in the introduction of the review. Now, the literature was added to 83 articles for a literature review, as followed. (Lines 484-704).

  • Li Y, Baptista R P, Kissinger J C. Noncoding RNAs in apicomplexan parasites: an update[J]. Trends in Parasitology, 2020, 36(10): 835-849.
  • Knopman D S, Amieva H, Petersen R C, et al. Alzheimer disease[J]. Nature reviews Disease primers, 2021, 7(1): 1-21.
  • Andersen J K. Oxidative stress in neurodegeneration: cause or consequence[J]. Nature medicine, 2004, 10(7): S18-S25.
  • Lima J A, Hamerski L. Alkaloids as potential multi-target drugs to treat Alzheimer's disease[J]. Studies in natural products chemistry, 2019, 61: 301-334.
  • Sharma C, Kim S, Nam Y, et al. Mitochondrial dysfunction as a driver of cognitive impairment in Alzheimer’s disease[J]. International Journal of Molecular Sciences, 2021, 22(9): 4850.
  • Von Bernhardi R, Eugenín J. Alzheimer's disease: redox dysregulation as a common denominator for diverse pathogenic mechanisms[J]. Antioxidants & redox signaling, 2012, 16(9): 974-1031.
  • Gitler A D, Dhillon P, Shorter J. Neurodegenerative disease: models, mechanisms, and a new hope[J]. Disease models & mechanisms, 2017, 10(5): 499-502.
  • Farkhondeh T, Forouzanfar F, Roshanravan B, et al. Curcumin effect on non-amyloidogenic pathway for preventing alzheimer’s disease[J]. Biointerface Research in Applied Chemistry, 2019, 9(4): 4085-4089.
  • Bălașa A F, Chircov C, Grumezescu A M. Body Fluid Biomarkers for Alzheimer’s Disease—An Up-To-Date Overview[J]. Biomedicines, 2020, 8(10): 421.
  • Khalil M, Teunissen C E, Otto M, et al. Neurofilaments as biomarkers in neurological disorders[J]. Nature Reviews Neurology, 2018, 14(10): 577-589.
  • Battaglia S, Garofalo S, di Pellegrino G. Context-dependent extinction of threat memories: influences of healthy aging[J]. Scientific reports, 2018, 8(1): 1-13.
  • Serafín V, Gamella M, Pedrero M, et al. Enlightening the advancements in electrochemical bioanalysis for the diagnosis of Alzheimer’s disease and other neurodegenerative disorders[J]. Journal of Pharmaceutical and Biomedical Analysis, 2020, 189: 113437.
  • Nazam F, Shaikh S, Nazam N, et al. Mechanistic insights into the pathogenesis of neurodegenerative diseases: towards the development of effective therapy[J]. Molecular and Cellular Biochemistry, 2021, 476(7): 2739-2752.
  • Cui Y, Yang M, Zhu J, et al. Developments in diagnostic applications of saliva in Human Organ Diseases[J]. Medicine in Novel Technology and Devices, 2022: 100115.
  • Brito-Aguilar R. Dementia around the world and the Latin America and Mexican scenarios[J]. Journal of Alzheimer's Disease, 2019, 71(1): 1-5.
  • Liang D, Lu H. Salivary biological biomarkers for Alzheimer’s disease[J]. Archives of oral biology, 2019, 105: 5-12.
  • Mollan S P, Davies B, Silver N C, et al. Idiopathic intracranial hypertension: consensus guidelines on management[J]. Journal of Neurology, Neurosurgery & Psychiatry, 2018, 89(10): 1088-1100.
  • Chávez-Gutiérrez L, Szaruga M. Mechanisms of neurodegeneration—Insights from familial Alzheimer’s disease[C]//Seminars in Cell & Developmental Biology. Academic Press, 2020, 105: 75-85.
  • Brazaca L C, Sampaio I, Zucolotto V, et al. Applications of biosensors in Alzheimer's disease diagnosis[J]. Talanta, 2020, 210: 120644.
  • Yao F, Zhang K, Zhang Y, et al. Identification of blood biomarkers for Alzheimer's disease through computational prediction and experimental validation[J]. Frontiers in neurology, 2019, 9: 1158.
  • Jack Jr C R, Bennett D A, Blennow K, et al. NIA‐AA research framework: toward a biological definition of Alzheimer's disease[J]. Alzheimer's & Dementia, 2018, 14(4): 535-562.
  • Tanaka M, Toldi J, Vécsei L. Exploring the etiological links behind neurodegenerative diseases: Inflammatory cytokines and bioactive kynurenines[J]. International Journal of Molecular Sciences, 2020, 21(7): 2431.
  • Lake J, Storm C S, Makarious M B, et al. Genetic and Transcriptomic Biomarkers in Neurodegenerative Diseases: Current Situation and the Road Ahead[J]. Cells, 2021, 10(5): 1030.
  • Kim K Y, Shin K Y, Chang K A. Brain-Derived Exosomal Proteins as Effective Biomarkers for Alzheimer’s Disease: A Systematic Review and Meta-Analysis[J]. Biomolecules, 2021, 11(7): 980.
  • Motataianu A, Barcutean L I, Maier S, et al. Cardiac Autonomic Neuropathy in Diabetes Mellitus Patients-Are We Aware of the Consequences[J]. Acta Medica Marisiensis, 2020, 66(1):1-7.
  • Leandrou S, Lamnisos D, Mamais I, et al. Assessment of Alzheimer’s disease based on texture analysis of the entorhinal cortex[J]. Frontiers in Aging Neuroscience, 2020, 12: 176.
  • Suppiah S, Didier M A, Vinjamuri S. The who, when, why, and how of PET amyloid imaging in management of Alzheimer’s disease—Review of literature and interesting images[J]. Diagnostics, 2019, 9(2): 65.
  • Monajjemi M. Molecular vibration of dopamine neurotransmitter: A relation between its normal modes and harmonic notes[J]. Biointerface Res. Appl. Chem, 2019, 9: 3956-3962.
  • Gupta J, Fatima M T, Islam Z, et al. Nanoparticle formulations in the diagnosis and therapy of Alzheimer's disease[J]. International journal of biological macromolecules, 2019, 130: 515-526.
  • Montgomery W, Goren A, Kahle-Wrobleski K, et al. Detection, diagnosis, and treatment of Alzheimer’s disease dementia stratified by severity as reported by caregivers in Japan[J]. Neuropsychiatric Disease and Treatment, 2018, 14: 1843.
  • Kim J, Campbell A S, de Ávila B E F, et al. Wearable biosensors for healthcare monitoring[J]. Nature biotechnology, 2019, 37(4): 389-406.
  • Farah R, Haraty H, Salame Z, et al. Salivary biomarkers for the diagnosis and monitoring of neurological diseases[J]. Biomedical journal, 2018, 41(2): 63-87.
  • Jasim H, Olausson P, Hedenberg-Magnusson B, et al. The proteomic profile of whole and glandular saliva in healthy pain-free subjects[J]. Scientific reports, 2016, 6(1): 1-10.
  • Jasim H, Carlsson A, Hedenberg-Magnusson B, et al. Saliva as a medium to detect and measure biomarkers related to pain[J]. Scientific reports, 2018, 8(1): 1-9.
  • Lee M, Guo J P, Kennedy K, et al. A method for diagnosing Alzheimer’s disease based on salivary amyloid-β protein 42 levels[J]. Journal of Alzheimer's Disease, 2017, 55(3): 1175-1182.
  • Ashton N J, Ide M, Schöll M, et al. No association of salivary total tau concentration with Alzheimer's disease[J]. Neurobiology of aging, 2018, 70: 125-127.
  • Kodintsev A N, Kovtun O P, Volkova L I. Saliva biomarkers in diagnostics of early stages of Alzheimer's disease[J]. Neurochemical Journal, 2020, 14(4):429-438.
  • 38[16] Li Y, Li R, Li X, et al. Effects of different aerobic exercise training on glycemia in patients with type 2 diabetes: A protocol for systematic review and meta-analysis [J]. Medicine, 2021, 100(18): e25615.
  • Almeida Pdel V, Gregio A M, Machado M A, et al. Saliva composition and functions: a comprehensive review[J]. The journal of contemporary dental practice,2008, 9:72–80.
  • Lee, Kwon, Shin, et al. Optimization of Saliva Collection and Immunochromatographic Detection of Salivary Pepsin for Point-of-Care Testing of Laryngopharyngeal Reflux[J]. Sensors, 2020, 20(1):325-333.
  • Kai D T, Kenny L, Frazer I H, et al. High‐risk human papillomavirus detection in oropharyngeal cancers: Comparison of saliva sampling methods[J]. Head & Neck, 2019, 41: 1484-1489.
  • Kara D, Bayrak N A, Volkan B, et al. Anxiety and salivary cortisol levels in children undergoing esophago-gastro-duodenoscopy under sedation[J]. Journal of pediatric gastroenterology and nutrition, 2019, 68(1): 3-6.
  • Cui Y, Zhang H, Zhu J, et al. Unstimulated Parotid Saliva Is a Better Method for Blood Glucose Prediction[J]. Applied Sciences, 2021, 11(23): 11367.
  • Cui Y, Zhang H, Zhu J, et al. Correlations of Salivary and Blood Glucose Levels among Six Saliva Collection Methods[J]. International Journal of Environmental Research and Public Health, 2022, 19(7): 4122.
  • Blennow K, Hampel H. CSF markers for incipient Alzheimer's disease[J]. The Lancet Neurology, 2003, 2(10): 605-613.
  • Buerger K, Ewers M, Pirttilä T, et al. CSF phosphorylated tau protein correlates with neocortical neurofibrillary pathology in Alzheimer's disease[J]. Brain, 2006, 129(11): 3035-3041.
  • Yousif T I, O’Reilly K, Nadeem M. Blood tests are not always helpful in predicting bacterial meningitis in children[J]. Sudanese journal of paediatrics, 2016, 16(2): 77.
  • Bellagambi F G, Lomonaco T, Salvo P, et al. Saliva sampling: Methods and devices. An overview[J]. TrAC Trends in Analytical Chemistry, 2020, 124: 115781.
  • Gupta S, Ahuja N. Salivary glands[J]. London, United Kingdom: IntechOpen, 2019: 63-76.
  • Pedersen A M L, Sørensen C E, Proctor G B, et al. Salivary secretion in health and disease[J]. Journal of oral rehabilitation, 2018, 45(9): 730-746.
  • Mohamed R, Campbell J L, Cooper-White J, et al. The impact of saliva collection and processing methods on CRP, IgE, and Myoglobin immunoassays[J]. Clinical and translational medicine, 2012, 1(1): 1-8.
  • Hernndez L M, Taylor M K. Salivary gland anatomy and physiology[J]. Salivary bioscience: foundations of interdisciplinary saliva research and applications. New York: Springer Nature, 2020: 11-20.
  • Punyadeera C. Human saliva as a tool to investigate intimate partner violence[J]. Brain, behavior, and immunity, 2012, 26(4): 541-542.
  • Niedrig M, Patel P, Abd El Wahed A, et al. Find the right sample: A study on the versatility of saliva and urine samples for the diagnosis of emerging viruses[J]. BMC infectious diseases, 2018, 18(1): 1-14.
  • Hartenbach F A R R, Velasquez É, Nogueira F C S, et al. Proteomic analysis of whole saliva in chronic periodontitis[J]. Journal of proteomics, 2020, 213: 103602.
  • Xin H, Katakowski M, Wang F, et al. MicroRNA-17–92 cluster in exosomes enhance neuroplasticity and functional recovery after stroke in rats[J]. Stroke, 2017, 48(3): 747-753.
  • Müller U C, Deller T, Korte M. Not just amyloid: physiological functions of the amyloid precursor protein family[J]. Nature Reviews Neuroscience, 2017, 18(5): 281-298.
  • Farah R, Haraty H, Salame Z, et al. Salivary biomarkers for the diagnosis and monitoring of neurological diseases[J]. Biomedical journal, 2018, 41(2): 63-87.
  • Ashton N J, Ide M, Schöll M, et al. No association of salivary total tau concentration with Alzheimer's disease[J]. Neurobiology of aging, 2018, 70: 125-127.
  • Sabbagh M N, Shi J, Lee M, et al. Salivary beta amyloid protein levels are detectable and differentiate patients with Alzheimer’s disease dementia from normal controls: Preliminary findings[J]. BMC neurology, 2018, 18(1): 1-4.
  • Blennow K, Hampel H, Weiner M, et al. Cerebrospinal fluid and plasma biomarkers in Alzheimer disease[J]. Nature Reviews Neurology, 2010, 6(3): 131-144.
  • Pekeles H, Qureshi H Y, Paudel H K, et al. Development and validation of a salivary tau biomarker in Alzheimer's disease[J]. Alzheimer's & Dementia: Diagnosis, Assessment & Disease Monitoring, 2019, 11: 53-60.
  • Reale M, Gonzales-Portillo I, Borlongan C V. Saliva, an easily accessible fluid as diagnostic tool and potent stem cell source for Alzheimer’s Disease: Present and future applications[J]. Brain Research, 2020, 1727: 146535.
  • Chandra A. Role of amyloid from a multiple sclerosis perspective: a literature review[J]. Neuroimmunomodulation, 2015, 22(6): 343-346.
  • Holmberg B, Johnels B, Blennow K, et al. Cerebrospinal fluid Aβ42 is reduced in multiple system atrophy but normal in Parkinson's disease and progressive supranuclear palsy[J]. Movement disorders: official journal of the Movement Disorder Society, 2003, 18(2): 186-190.
  • Lewczuk P, Lelental N, Spitzer P, et al. Amyloid-β 42/40 cerebrospinal fluid concentration ratio in the diagnostics of Alzheimer's disease: validation of two novel assays[J]. Journal of Alzheimer's disease, 2015, 43(1): 183-191.
  • Albahri A S, Alwan J K, Taha Z K, et al. IoT-based telemedicine for disease prevention and health promotion: State-of-the-Art[J]. Journal of Network and Computer Applications, 2021, 173: 102873.
  • Pekeles H, Qureshi H Y, Paudel H K, et al. Development and validation of a salivary tau biomarker in Alzheimer's disease[J]. Alzheimer's & Dementia: Diagnosis, Assessment & Disease Monitoring, 2019, 11: 53-60.
  • Park S A, Han S M, Kim C E. New fluid biomarkers tracking non-amyloid-β and non-tau pathology in Alzheimer’s disease[J]. Experimental & molecular medicine, 2020, 52(4): 556-568.
  • Ebneth A, Godemann R, Stamer K, et al. Overexpression of tau protein inhibits kinesin-dependent trafficking of vesicles, mitochondria, and endoplasmic reticulum: implications for Alzheimer's disease[J]. The Journal of cell biology, 1998, 143(3): 777-794.
  • Kent S A, Spires-Jones T L, Durrant C S. The physiological roles of tau and Aβ: implications for Alzheimer’s disease pathology and therapeutics[J]. Acta neuropathologica, 2020, 140(4): 417-447.
  • Iqbal K, Liu F, Gong C X. Tau and neurodegenerative disease: the story so far[J]. Nature reviews neurology, 2016, 12(1): 15-27.
  • Lau H C, Lee I K, Ko P W, et al. Non-invasive screening for Alzheimer’s disease by sensing salivary sugar using Drosophila cells expressing gustatory receptor (Gr5a) immobilized on an extended gate ion-sensitive field-effect transistor (EG-ISFET) biosensor[J]. PLoS One, 2015, 10(2): e0117810.
  • Shi M, Sui Y T, Peskind E R, et al. Salivary tau species are potential biomarkers of Alzheimer's disease[J]. Journal of Alzheimer's Disease, 2011, 27(2): 299-305.75.
  • Zetterberg H, Blennow K. Moving fluid biomarkers for Alzheimer’s disease from research tools to routine clinical diagnostics[J]. Molecular neurodegeneration, 2021, 16(1): 1-7.
  • Cano A, Turowski P, Ettcheto M, et al. Nanomedicine-based technologies and novel biomarkers for the diagnosis and treatment of Alzheimer’s disease: from current to future challenges[J]. Journal of Nanobiotechnology, 2021, 19(1): 1-30.
  • Lee J C, Kim S J, Hong S, et al. Diagnosis of Alzheimer’s disease utilizing amyloid and tau as fluid biomarkers[J]. Experimental & molecular medicine, 2019, 51(5): 1-10.
  • Pawlowski M, Meuth S G, Duning T. Cerebrospinal fluid biomarkers in Alzheimer’s disease—From brain starch to bench and bedside[J]. Diagnostics, 2017, 7(3): 42.
  • Niemantsverdriet E, Valckx S, Bjerke M, et al. Alzheimer’s disease CSF biomarkers: Clinical indications and rational use[J]. Acta neurologica belgica, 2017, 117(3): 591-602.
  • Vogel J W, Iturria-Medina Y, Strandberg O T, et al. Spread of pathological tau proteins through communicating neurons in human Alzheimer’s disease[J]. Nature communications, 2020, 11(1): 1-15.
  • Jin M, Cao L, Dai Y. Role of neurofilament light chain as a potential biomarker for Alzheimer's disease: a correlative meta-analysis[J]. Frontiers in Aging Neuroscience, 2019, 11: 254.
  • Battaglia S, Harrison B J, Fullana M A. Does the human ventromedial prefrontal cortex support fear learning, fear extinction or both? A commentary on subregional contributions[J]. Molecular Psychiatry, 2021: 1-3.
  • Tanaka M, Vécsei L. Editorial of Special Issue “Crosstalk between depression, anxiety, and dementia: comorbidity in behavioral neurology and neuropsychiatry [J]. Biomedicines, 2021, 9(5): 517.

Point 4: Introduction: The ‘Introduction’ section is well-written and nicely presented, with a good balance of information about molecular pathophysiology and neurological changes associated with Alzheimer’s disease. Nevertheless, I believe that more information about structural and functional changes associated with neurodegeneration in AD will provide a better and more accurate background. Thus, I suggest the authors to make such an effort to provide a brief overview of the pertinent published on neurobiological signs of AD, because as it stands, this information is not highlighted in the text. In this regard, I believe that the statement ‘…neuropathologic markers of AD may manifest ten to fifteen years before evident cognitive symptoms, which can be characterized as substantial memory concern, early mild cognitive impairment (MCI), or late MCI’ needs some necessary citations. In particular, according with this sentence, I would recommend citing a recent review that examined pathophysiological basis and biomarkers of AD pathology and investigated molecular signs of neuroinflammation in neurodegenerative diseases, in particular Alzheimer’s disease (https://doi.org/10.3390/ijms21072431). Importantly, I also recommend a relevant study in which author investigated age-related impairments in the ability to process contextual information and in the regulation of responses to threat, addressing that structural and physiological alterations in the prefrontal cortex and medial temporal lobe determine cognitive changes in advanced aging, that can eventually cause patterns of cognitive dysfunctions observed in patients with AD/MCI (https://doi.org/10.1038/s41598-018-31000-9). I firmly believe that these improvements will help to provide a more coherent and defined background.

Response 4: Thank you for your careful review and valuable suggestions. As the reviewer stated, we have restructured the Introduction section and added the section mentioned in Point 4, as followed. (Lines 35-59, Lines 69-75 and Lines 85-95).

Neurodegeneration is a multidimensional process involving multiple biochemical pathways and a complex interplay of a range of regulatory variables [1,2]. It is defined by the progressive and irreversible loss of neurons from certain brain and spinal cord regions, most notably the nuclei of the base within subcortical areas and the cerebral cortex, resulting in damage and dysfunction exhibited as cognitive and motor dysfunctions [3,4]. In general, the causative factors include oxidative stress and free radical formation; protein misfolding, oligomerization, and aggregation; mitochondrial dysfunction, axonal transport deficits, and abnormal neuron–glial interactions; calcium deregulation and phosphorylation impairment; neuroinflammation; DNA damage and aberrant RNA processing [5,6].

Numerous chronic and incurable age-related illnesses are caused by neurodegeneration [7]. The frequency of neurodegenerative illnesses is steadily increasing as a result of technological advancements and growth, posing a substantial threat to human health [8]. Additionally, neurodegeneration is associated with a variety of neurodegenerative, neurotraumatic, and neuropsychiatric disorders, each of which has a significantly different pathophysiology, including memory and cognitive impairments, muscle weakness and/or paralysis, abnormal voluntary movement control, seizures, confusion, and pain [9]. Such diseases range in severity from progressive degenerative disorders such as Alzheimer's disease (AD), Parkinson's disease, Huntington's disease, amyotrophic lateral sclerosis, and multiple sclerosis to acute traumatic injuries such as traumatic brain injury, stroke, or spinal cord injury [10,11].

AD is the most prevalent neurodegenerative condition, affecting about 45 million people globally and estimated to reach 60 million by 2030 as the elderly population grows [12]. It is the most common cause of dementia in late adulthood and is associated with a substantial socioeconomic cost, as well as increased morbidity and mortality. However, no effective treatment strategy for AD is currently available [13]. The primary risk factor for the majority of neurodegenerative disorders, including AD, is advancing age [14]. According to the WHO, dementia is a fast-expanding public health problem, affecting approximately 50 million people worldwide in 2019 [15], with AD accounting for 60% to 70% of cases. Due to the constraints of early AD diagnosis and frequent therapy delays, the therapeutic impact would gradually deteriorate. Although lumbar puncture is the most often used procedure for diagnosing AD, its invasiveness can cause pain to the patient and is difficult for many individuals to accept [16,17]. AD is characterized by the progressive death of cholinergic neurons in the hippocampus and cortex, resulting in atrophy, neurotransmission anomalies and synaptic loss, and neurodegeneration [18]. The basic causes of AD entail the extracellular deposition of amyloid beta (Aβ) peptides and hyperphosphorylated tau protein aggregates. Intracellular production of neurofibrillary tangles that cause oxidative stress, persistent neuroinflammation, neuronal dysfunction, and neurodegeneration [19,20]. Jack et al. [21] discovered that the neuropathologic markers of AD may manifest ten to fifteen years before evident cognitive symptoms, which can be characterized as substantial memory concern, early mild cognitive impairment (MCI), or late MCI. It is possible to delay the onset of AD, alleviate patient suffering, and alleviate social burdens through early diagnosis and treatments [22]. The most recent guidelines stated unequivocally that biomarkers should be used as indicators for clinical diagnosis of AD, particularly in the early stages, and that this was almost the only way to do so [23,24]. As a result, it is critical to develop highly sensitive and specific biomarkers as well as more reliable and minimally intrusive methods for early detection of AD.

Current diagnostic procedures, in addition to cognitive testing, rely on imaging techniques [25] and cerebrospinal fluid (CSF) measurements. On the one hand, neuroimaging techniques use magnetic resonance imaging to assess hippocampal atrophy and positron emission tomography to assess cortical A deposition. Cerebrospinal fluid analysis, on the other hand, strives to offer quantitative assessments of Aβ and tau protein levels as biomarkers for AD [26,27]. Existing approaches, on the other hand, are costly, rather invasive [28], and have low sensitivity and specificity, posing hazards of overdiagnosis or underdiagnosis, misattribution, or omission and neglect of symptoms [29]. Furthermore, due to a chronic paucity of AD diagnostic testing across all disease stages, individuals are frequently detected late, imposing a significant burden on health-care systems [30]. Saliva is a fluid that can be collected easily and noninvasively, and has been recommended as a source of readily available biomarkers for the diagnosis and evaluation of a variety of pathological disorders not only in the oral cavity, but also throughout the body [14,31]. Saliva glands produce saliva primarily in response to autonomic nervous system monitoring via cholinergic innervation of cranial nerves VII and IX [32]. These glandular secretions may reflect a variety of aspects of the nervous system's physiology. Indeed, proteins from the central nervous system are released in an age-dependent way [33]. Additionally, proteins can enter saliva from the blood by passive diffusion, active transport, or microfiltration [34]. As a result of these findings, saliva may contain novel indicators of central nervous system impairment, making it a more convenient and accessible source for capturing AD-related biomarkers. Recent research indicate that saliva may be a source of noninvasive indicators for AD diagnosis. These indicators may be produced directly in salivary glands or may diffuse from blood. Additionally, salivary levels may indicate changes in CSF. Several publications have detected and quantified salivary A40, A42, t-tau, and p-tau levels in patients with AD and controls [35,36].

Point 5: Participants: I suggest the authors to reshape this paragraph because, as it is, it may be difficult for the reader to fully comprehend that the first 15 controls were used to select the best salive collection method.

Response 5: Thank you for your careful review and valuable suggestions. In order to be easy for the reader to fully comprehend that the first 15 controls were used to select the best salive collection method. The paragraph in Participants section was reshaped as followed. (Lines 142-144 and Lines 150-153).

In this study, first, 15 healthy participants (8 men and 7 women), with a mean ± SD age of 67.5 ±3.2 years old were included. Each participant was collected saliva for 7 consecutive days with 6 different collection methods, and the mean value was utilized to determine the most optimal saliva collection method. Second, 30 patients with AD diagnosed were included. Selection criteria [42]: the diagnosis complied with the relevant standards formulated in "Chinese Mental Illness Classification Scheme and Diagnostic Criteria (CCMD-3)" and "Diagnosis and Statistics Manual of Mental Illness". Besides, age ≥ 60 years old. Exclusion criteria: severe organ dysfunction, took antipsychotic drugs a month before the experiment. Another 30 non-AD controls from the physical examination center during the same period were selected. Selection criteria: No mental illness, age ≥ 60 years old, no mental disorder or hereditary neurological disease among immediate family members, no history of head trauma and normal cognitive function.

Point 6: Laboratory test: I suggest the authors to rewrite this paragraph, because as it stands, this section is written with an informal style, and appears to be far too much impersonal and dispersive.

Response 6: Thank you for your careful review and valuable suggestions. This section was rewritten with a formal style, in order to do not appear to be far too much impersonal and dispersive as followed. (Lines 155-177).

30 minutes before collection, refrain from smoking, cleaning teeth, eating, or drinking. After that, the mouth was washed with water to remove any food remnants in the oral cavity [43]. Saliva was collected using a salivette (Sarstedt, 51.5134) (containing untreated swabs and swabs activated by citric acid), as well as six other collection methods [44].

With and without stimulation, samples of the parotid gland, mandibular/sublingual gland, and total saliva were taken from each participant. Unstimulated whole saliva (UWS), stimulated whole saliva (SWS), unstimulated sublingual/submandibular saliva (USS), stimulated sublingual/submandibular saliva (SSS), stimulated parotid saliva (SPS), unstimulated parotid saliva (SPS), stimulated parotid saliva (SPS), stimulated parotid saliva (SPS), stimulated parotid saliva (SPS (UPS). The collection time was between 9:00 and 9:30 a.m. In the same clinical room, all saliva samples were taken in the same order. To prevent delicate peptides from degrading, all samples were collected in pre-chilled polypropylene tubes on ice. The total amount of saliva obtained by all methods was 5 mL. Finally, it was transferred to the laboratory on a regular basis and centrifuged at -20°C for subsequent use.

The six saliva collection methods were evaluated in the morning on 15 healthy subjects in order to determine the optimal saliva collection method. The chosen procedures were then applied to 30 AD patients and 30 non-AD controls. Salivary Aβ40, Aβ42, t-tau and p-tau levels were determined using a particular enzyme-linked immunosorbent test (ELISA, Beijing Furui Runkang Biotechnology Co., Ltd., China). The saliva ELISA kit showed a sensitivity of 0.537 ng/mL, a variation of 2.6% within the assay, and a variation of 6.6% between the assays.

Point 7 : Results: In my opinion, this section is well organized, but it illustrates findings in an excessively broad way, without really providing full statistical details, to ensure in-depth understanding and replicability of the findings. Indeed, I believe that it is necessary for the authors to present their findings with a precise description not just in the in descriptive tables, but also in the main text.

Response 7: Thank you for your careful review and valuable suggestions. We present our findings with a precise description not only in the in descriptive tables, but also in the main text as followed. (Lines 205-207 and Lines 225-230).

Fig. 1 was the summary of saliva data. It showed the salivary Aβ40, Aβ42, t-tau, p-tau, Aβ42/Aβ40 and t-tau/p-tau concentration at different saliva collection methods. It could be seen that the salivary Aβ42 and t-tau concentration of UPS were significantly higher than the other five methods. At the same time, in salivary Aβ40, Aβ42, and t-tau the stimulated saliva sample concentration was significantly lower than the unstimulated saliva sample concentration. However, the opposite conclusion was obtained in salivary p-tau. The maximum value of Aβ40/Aβ42 was obtained in UPS, however, the maximum value of p-tau/t-tau was obtained in SSS.

This study included 60 subjects (30 women and 30 men), with an average age of 72.3 ± 5.2 (60-89 years), 30 patients (50%) and 30 controls (50%). There was no statistical difference in age or gender between the two groups. Table 2 described the saliva analytes of the study group and the results of univariate and multivariate logistic regression models regarding the effects of parameters on patients with dementia. In univariate analysis, individuals with higher Aβ42 levels were more likely to belong to AD group. In the multivariate regression analysis, the levels of Aβ42/Aβ40 and p-tau/t-tau were similar, but the combination of p-tau and t-tau was better than a single factor with statistically significant.

Point 8: Discussion: In this final section, authors described the results and their argumentation and captured the state of the art well; however, I would have liked to see some views on a way forward. I believe that the authors should make an effort, trying to explain the theoretical implication as well as the translational application of this research article, to adequately convey what they believe is the take-home message of their study. Discussion of theoretical and methodological avenues in need of refinement is necessary, as well as suggestions of a path forward in the understanding of supportive interventions and treatment for people with dementia. In particular, since the focus of this research was on Aβ, t-tau and p-tau levels in AD, I would suggest adding some studies that might discuss amyloid-β (Aβ) pathology in AD, highlighting the combined effect of forms of Aβ and tau protein to drive healthy neurons into the diseased state. Aβ peptide and tau protein consistently accumulate in the frontal and/or parietal lobes, and cause alterations of frontal lobe that impact memory and error-driven learning in individuals who have a high risk of dementia: evidence from a recent theoretical review (https://doi.org/10.1038/s41380-021-01326-4) that focused on neurobiology of fear conditioning, analyzed the role of ventromedial prefrontal cortex (vmPFC) in the processing of safety-threat information and their relative value, and how this region is fundamental for the evaluation and representation of stimulus-outcome’s value needed to produce sustained physiological responses. Also, I believe that a recent yet relevant perspective manuscript (https://doi.org/10.17219/acem/146756) might be of interest: here the focus was on providing a deeper understanding of human learning neural networks, particularly on human PFC crucial role, that might also contribute to the advancement of alternative, more precise and individualized treatments for psychiatric disorders. Secondary, authors also might to consider some studies that have focused on this topic (https://doi.org/10.3390/biomedicines10010076; https://doi.org/10.3390/biomedicines9050517). I believe that adding information from these studies may improve the interpretation of the results of the present article and their argumentation by highlighting how cognitive alterations caused by frontal dysfunction are fundamental as neurodegenerative biomarkers of AD.

Response 8: Thank you for your careful review and valuable suggestions. As the reviewer stated, we have restructured the Discussion section and added the section mentioned in Point 4, as followed. (Lines 304-309, Lines 326-340, Lines 350-367, Lines 382-391 and Lines 405-453).

The detection of AD from an early phase characterized by mild cognitive deterioration is very important, since only an early diagnosis can afford long-term symptom relief. Techniques capable of detecting the disorder in its early stages are thus urgently needed [45]. In this study, six saliva collection methods were employed in 15 healthy participants in the morning to choose the ideal saliva collection method. Then the chosen methods in different biomarkers were used of 30 AD patients and 30 non-AD controls. ELISA was used to determine the levels of salivary Aβ40, Aβ42, t-tau, and p-tau of these participants. It was particularly important that we could detect the levels of salivary Aβ40, Aβ42, t-tau and p-tau through a simple and reproducible method. This study also demonstrated the effectiveness of saliva as a diagnostic biological fluid depended on the standardization of collection methods to deliver the most accurate and meaningful results. Different saliva collection methods have a great influence on the concentration and correlation of Aβ40, Aβ42, t-tau and p-tau. Therefore, the standardization of saliva collection method is pivotal to minimizing the effect on the variations in saliva composition within and between individuals. The CSF biomarkers of AD reflect the key factors of the physiopathology of the disease and provide 90-95% sensitivity and specificity [46]. However, obtaining samples from CSF examination is a highly invasive and stressful process. For this reason, search for less invasive methods to diagnose and monitor AD is gaining attention. Compared with CSF, blood measurements are advantageous for AD biomarker screening because blood collection is easier and less invasive. However, drawing blood can make some people squeamish. Ranging from a slight discomfort to creating a panicked and fearful state, some people would rather not get blood work taken. Blood tests are not always definitive. Sometimes instead of providing a solution and answer they instead raise more questions. Besides, waiting for the results of blood test can generate a great amount of anxiety [47]. On the other hand, saliva sampling is noninvasive, safe, cheap, and easily performed compared to blood [48]. This fact makes it a biological fluid for the research and monitoring of biomarkers for various diseases. However, there are few studies of the behavior of saliva biomarkers and their utility in AD.

Saliva is produced by three major salivary glands (parotid, submandibular and sublingual) and numerous minor salivary glands. Saliva contains a myriad of salivary proteins which could serve as biological markers for diagnosing and tracking the progression of various health conditions, as well as monitoring the effectiveness of medication [49,50]. To date, most of the saliva collection devices that are commercially available allow a person to collect resting/unstimulated saliva and/or stimulated saliva either via mechanical stimulation or acid stimulation [51]. When a person is in a resting state, saliva production is largely produced by the submandibular gland, while only 20% and 8% are produced by parotid and sublingual glands, respectively [52]. In contrast, when saliva production is stimulated through acid stimulation, most of the saliva produced is primarily derived from the parotid gland [50]. Most importantly, the composition of both stimulated and unstimulated saliva may be altered by genetic predisposition factors and physiological, pathological and environmental factors [53], all these factors may hinder the correct derivation of results for best care outcomes. Therefore, this study determined the possible optimal collection method for each marker through the collection and detection of saliva markers from 15 elderly healthy individuals for 7 consecutive days.

The relative contribution of different glands to the whole saliva varies with the collection method, the degree of stimulation, age and even the time of day [34]. The variable nature of saliva secretion suggests that different methods may have to be used when studying its components or their possible role as indicators of specific physiological conditions. There is a large amount of literature on the diagnostic possibilities of saliva, but there is still no standardized method for collecting saliva samples. In different studies, different sampling methods are often used, and many studies do not or rarely describe patient preparation or sampling procedures [54]. In addition, without proper clinical examination, the characteristics of participants are usually insufficient. Most saliva Aβ40, Aβ42, t-tau and p-tau research papers focus on studying the whole saliva [55] because it can be easily obtained by spitting it into a test tube or letting it drip from the mouth. Few people pay attention to ductal saliva obtained from different salivary glands. Moreover, comparing the Aβ40, Aβ42, t-tau and p-tau expression of whole saliva and glandular saliva was focused on in a cohort of careful characterization and clinical examination. The results indicate that different collection methods provide significant differences in the snapshots of salivary Aβ40, Aβ42, t-tau and p-tau. In this study, the results indicated that different saliva collection methods provided significant differences in the snapshots of saliva biomarkers for AD. Based on the comparison of unstimulated and stimulated saliva collection methods, it can be shown that based on the simplicity and low variability of the collection method, different biomarkers may be preferred different collection methods. Therefore, standardization of saliva collection method is pivotal to minimising the effect on the variations in saliva composition within and between individuals. The alternative saliva collection methods (e. g. USS in Aβ40, UPS in Aβ42, UPS in t-tau, SSS in p-tau 181) would be an ideal way to collect saliva in a clinical, challenging environment. It was particularly important that we could detect the levels of salivary Aβ40, Aβ42, t-tau and p-tau through a simple and reproducible method. In the present study, we determined the levels of salivary Aβ40, Aβ42, t-tau and p-tau using six saliva collection methods included UWS, SWS, USS, SSS, SPS, UPS. Different saliva collection methods have a great influence on the concentration and correlation of Aβ40, Aβ42, t-tau and p-tau.

We found that compared with the control group, statistically significant increase levels of salivary Aβ42 in AD patients. In addition, according to the ROC analysis performed for the predictive value of Aβ42 levels, the AUC was 0.848. In addition, salivary Aβ40 expression was unchanged within all the studied sample. We also analyzed the ratio of salivary Aβ42/Aβ40 and we found that compared with the control group, AD patients were higher, but not statistically significant (p = 0.2). In general, compared with the control group, the level of salivary Aβ42 of early AD patients was significantly increased, which indicated that measuring salivary Aβ42 can be used as a biomarker to identify and confirm early AD diagnosis.

The miscleavage of amyloid precursor protein (APP), the pathological accumulation of which underpins AD, is the genesis of Aβ plaques. APP is cleaved into soluble APP by -secretase under physiological conditions, which is then cleaved into p3 peptide and APP intracellular domain by secretase [56]. APP has been established in studies to have a key function in brain homeostasis as well as neuronal growth and maturation during brain development [57]. In AD, APP is cleaved by beta-secretase and gamma-secretase rather than alpha-secretase. This enzymatic breakage cascade produces amyloid Aβ40 and Aβ42 peptides, which collect in the extracellular space and form plaques, causing neurotoxicity and activating reactive inflammatory processes, which eventually lead to neuronal damage [58]. This amyloidogenic pathway is a well-known source of diagnostic biomarkers for AD. Aβ deposits detectable by PET scans, and Aβ levels in cerebrospinal fluid, as well as other body fluids, are utilized as a diagnostic technique for AD The levels of Aβ40 and Aβ42 are the most reliable techniques for AD diagnosis among the numerous A subtypes. Aβ42 specifically accumulates form plaques in the brain, and its concentration in cerebrospinal fluid falls, which is a sign of AD Although Aβ40 was the most prevalent isoform, its levels in Alzheimer's patients did not decrease appreciably. As a result, our study concentrated on Aβ42 and Aβ40.

Therefore, it is particularly important to identify stable and reproducible salivary Aβ42 expression. Because it can be used as a potential indicator of AD neuropathology, it can be measured with the least stress on the subject. The mechanism of salivary Aβ42 accumulation is unclear. The localization may be due to the release of the peptide from the saliva gland due to the secretion of secretase from the saliva gland epithelial cells during the processing of amyloid precursor protein (APP) [59]. The results of this study are consistent with previous studies using CSF and plasma samples [60, 61]. The Aβ42 content of AD patients is approximately 2.12 times that of the control group. Studies have reported that Aβ42 is reduced in patients’ CSF [62], it can be understood that Aβ in the AD group is due to the damage of the blood brain barrier, which leads to the loss of neurons that produce APP, and the accumulation of Aβ in the brain and the reduction of Aβ clearance. Since the latter does not exist in the saliva glands, the Aβ produced by the APP glands may accumulate in the saliva [63].

There are various plausible causes for the greater agreement between amyloid PET and salivary Aβ42 when the Aβ42/Aβ40 ratio is used instead of the Aβ42/Aβ40 ratio. It is possible that non-AD subcortical lesions cause high salivary Aβ42 levels. Several studies [64,65] have discovered elevated amounts of salivary Aβ42 in multiple system atrophy and multiple sclerosis. Another possibility is that variances in overall Aβ production and clearance lead to interindividual variability in total salivary Aβ levels. This is reinforced by recent observations indicating Aβ42 associated with Aβ40 in saliva, even in healthy controls. As a result, when salivary Aβ42 was utilized to detect AD brain pathology, the Aβ40 ratio used may have an effect on interindividual variability in total Aβ levels [66].

We found that compared with the control group, no difference in salivary t-tau and p-tau (using T181) concentration was found between AD patients and non-AD controls, which was consistent with the results obtained by Albahri et al. [67]. We analyzed the ratio of salivary p-tau/t-tau and found that there were significant differences between AD patients and non-AD controls, which was consistent with the results obtained by Pekeles et al. [68]. Several explanations for changes in the p-tau/t-tau ratio levels in AD and controls are worth considering. A small number of people who are clinically diagnosed as possible AD may not have tau pathology themselves [69]. Another explanation may be that even for those AD subjects who showed elevated tau levels in the brain, some of these people still failed to express tau peripherally in the saliva gland tissue. Alternatively, it is possible that abnormal tau is present in most saliva tissues, but only in a subgroup of AD individuals secreted abnormal tau in saliva.

Tau proteins are members of the microtubule-associated protein family and, by binding to tubulin, contribute to microtubule stability and flexibility [70]. Tau phosphorylation normally promotes its disintegration from microtubules, as well as its instability and removal [71]. Mutations in the tau protein sequence have been shown to change its phosphorylation site, resulting in tau hyperphosphorylation [72]. p-tau can combine and form neurofibrillary tangles within cells. Lau et al. [73] evaluated t-tau and p-tau. The findings revealed that there were no quantitative differences between t-tau and p-tau patients in AD patients and healthy participants. However, in AD patients, there was a modest increase in p-tau levels (p< 0.05), which is consistent with the findings of this study. Min Shi et al. [74] investigated p-tau as a salivary biomarker for AD, and analysis of tau protein species, namely p-tau and t-tau, revealed that the p-tau/t-tau ratio of each individual was higher when compared to the healthy control group, which is consistent with the findings of this study. This shows that an increase in phosphorylated tau concentration relative to total tau concentration could be employed as a possible biomarker to detect AD [75].

Detection techniques based on novel biomarkers other than Aβ and tau protein may be a potential approach for the early detection of AD [76]. However, because no single biomarker can effectively diagnose AD, a combination of biomarkers can considerably enhance diagnostic accuracy [77, 78]. Although Aβ plaque deposition occurs years or even decades before symptom onset and can be utilized to make an early diagnosis, tau biomarkers alter with disease development and are closely linked to local degeneration and cognitive loss [79, 80]. Combining disease-specific and nonspecific biomarkers is the most effective technique for building biomarker-based diagnostic tools. In this context, the reduction in Aβ42, along with alterations in the Aβ42/Aβ40 ratio, t-tau, and p-tau levels, is commonly referred to as an Alzheimer's signature or signature because it allows for the detection of AD at an early stage [81]. Furthermore, their combined usage for AD diagnosis has a sensitivity and specificity of about 85-95% [80]. This is consistent with the study's result that when four markers are used together to predict AD, the accuracy is the highest (92.11%). As a result, combining biomarkers can enhance prediction accuracy.

However, our study has limitations. First of all, when studying several biomarkers, they have not been compared with gold standards such as CSF concentration and imaging. Although this study was conducted in patients who have already been diagnosed, the saliva biomarkers that reflect the current situation cannot be compared with image data obtained in some cases a few years ago. Therefore, the patient’s saliva biomarkers should be diagnosed when imaging or CSF is obtained for AD comparison to evaluate their salivary biomarkers. In addition, clinical studies have described the excellent diagnostic performance of saliva biomarkers. Although this study provides the best performance for early diagnosis of patients with cognitive problems and suspected AD, due to the small amount of data, further development is still needed, including validity testing, retesting, and multifactor testing. Finally, there are still many problems in this study that need to be resolved and further explored. For example, the submandibular glands and sublingual glands are closely located, so it is difficult to separate saliva from these glands with certainty, which was why saliva is collected from both glands in the current study. How to distinguish sublingual saliva from submandibular saliva is also a direction that needs further research. Finally, various risk factors (e. g. the role of ventromedial prefrontal cortex in the processing of safety-threat information and their relative value [82,83]) should also be included in the diagnosis of AD in saliva to make the diagnosis more accurate.

Point 9: In my opinion, I think the ‘Conclusions’ paragraph would benefit from some thoughtful as well as in-depth considerations by the authors, because as it stands, it is very descriptive but not enough theoretical as a discussion should be. Authors should make an effort, trying to explain the theoretical implication as well as the translational application of their research.

Response 9: Thank you for your careful review and valuable suggestions. In order to explain the theoretical implication as well as the translational application of their research. We restructured the ‘Conclusions’ paragraph as followed. (Lines 455-474).

In this study, the results indicated that different saliva collection methods provided significant differences in the snapshots of saliva biomarkers for AD. Based on the comparison of unstimulated and stimulated saliva collection methods, it can be shown that based on the simplicity and low variability of the collection method, different biomarkers may be preferred different collection methods. Therefore, standardization of saliva collection method is pivotal to minimising the effect on the variations in saliva composition within and between individuals. The alternative saliva collection methods (e. g. USS in Aβ40, UPS in Aβ42, UPS in t-tau, SSS in p-tau 181) would be an ideal way to collect saliva in a clinical, challenging environment. This study also showed that salivary biomarkers can be quantified and used to diagnose AD. We found significantly higher Aβ42 level in AD compared to control group. At the same time, Aβ40, t-tau, and p-tau did not change significantly, however, p-tau/t-tau and Aβ42/Aβ40 had a certain relevance. Among them, the AUC of 4 biomarkers combined in diagnosis of AD was large, equal to 92.11%.  In general, the results emphasize the importance of consistency when collecting saliva samples, which should be more important than the collection method itself. Our findings from this study pave the way towards making saliva diagnostics a reality for AD. With the further research and standardization of collection and quantification methods with larger sample groups, saliva biomarkers may become the gold standard for early diagnosis and screening of AD.

Point 10: In according to the previous comment, I would ask the authors to include a ‘Limitations and future directions’ section before the end of the manuscript, in which authors can describe in detail and report all the technical issues brought to the surface.

Response 10: Thank you for your careful review and valuable suggestions. In according to the previous comment, we added a ‘Limitations and future directions’ section before the end of the manuscript, we described in detail and report all the technical issues brought to the surface. (Lines 435-453).

However, our study has limitations. First of all, when studying several biomarkers, they have not been compared with gold standards such as CSF concentration and imaging. Although this study was conducted in patients who have already been diagnosed, the saliva biomarkers that reflect the current situation cannot be compared with image data obtained in some cases a few years ago. Therefore, the patient’s saliva biomarkers should be diagnosed when imaging or CSF is obtained for AD comparison to evaluate their salivary biomarkers. In addition, clinical studies have described the excellent diagnostic performance of saliva biomarkers. Although this study provides the best performance for early diagnosis of patients with cognitive problems and suspected AD, due to the small amount of data, further development is still needed, including validity testing, retesting, and multifactor testing. Finally, there are still many problems in this study that need to be resolved and further explored. For example, the submandibular glands and sublingual glands are closely located, so it is difficult to separate saliva from these glands with certainty, which was why saliva is collected from both glands in the current study. How to distinguish sublingual saliva from submandibular saliva is also a direction that needs further research. Finally, various risk factors (e. g. the role of ventromedial prefrontal cortex in the processing of safety-threat information and their relative value [82,83]) should also be included in the diagnosis of AD in saliva to make the diagnosis more accurate.

Point 11: Figures: I suggest to modify the Figures 2 and 4 for clarity because, as it stands, the readers may have difficulty comprehending it. In my opinion, authors should provide higher-resolution image of brain areas showed in the right panel, to allow a better detection of source regions of peak intensity. Also, please change the scale of the vertical axis and use the same minimum/maximum scale value in all the graphs.

Response 11: Thank you for your careful review and valuable suggestions. The limitations of our study are already in the last part of the manuscript, pointing out that one of the existing shortcomings is that we did not use cerebrospinal fluid contrast, and we did not obtain brain CT images, so we could not provide high-resolution images of brain regions proposed by the reviewers. But this does not affect the main purpose of our study

Salivary Aβ40, Aβ42, t-tau, and p-tau 181 were widely used in Alzheimer’s disease (AD) studies. However, saliva collection methods can influence the levels of these salivary biomarkers. This study aims to evaluate the effect of saliva collection methods on salivary Aβ40, Aβ42, t-tau, and p-tau 181 and identify the ideal saliva collection method. Because the effect of different collection methods on the content of saliva substances is very large and affects the use of saliva for disease diagnosis results, this study focuses on the effect of collection methods on the content of saliva substances, so it is not the material and methods of the actual research paper or in the supplementary section. At the same time, our main goal is to compare the effect of different collection methods on the content of substances in saliva, as well as to emphasize the importance of collection methods, so we did not compare with other substances, but this will not affect the overall results of the study. In this study, six saliva collection methods were employed in 15 healthy participants in the morning to choose the ideal saliva collection method. Then the chosen methods in different biomarkers were used of 30 AD patients and 30 non-AD controls. The levels of salivary Aβ42, Aβ40, t-tau and p-tau 181 were calculated by specific enzyme-linked immunosorbent assay (ELISA). The receiver operating characteristic was used to evaluate the value of saliva biomarkers in clinically diagnosed AD. The results showed that the highest levels of salivary Aβ40, Aβ42, t-tau, and p-tau were in different saliva collection methods. The unstimulated sublingual/submandibular saliva (USS) in Aβ40 had highest levels. Similarly, unstimulated parotid saliva (UPS) in Aβ42, unstimulated parotid saliva (UPS) in t-tau, and stimulated sublingual/submandibular saliva (SSS) in p-tau 181. The correlations between different saliva biomarkers in the same collection method were different. There was no significant correlation between Aβ40, Aβ42, t-tau and p-tau. Salivary Aβ42 was significantly higher in AD compared to non-AD controls. Meanwhile, Aβ40, t-tau, and p-tau did not change significantly, however, p-tau/t-tau and Aβ42/Aβ40 had a certain relevance. Among them, the area under the curve of 4 biomarkers combined in diagnosis of AD was the largest, equal to 92.11%. This study demonstrated that alternative saliva collection methods (e. g. USS in Aβ40, UPS in Aβ42, UPS in t-tau, SSS in p-tau 181) would be an ideal way to collect saliva in a clinical. Besides, combination of multiple biomarkers makes Alzheimer's diagnosis more precise. So, our conclusion is convincing. Finally, we have optimized the article again this time, the grammar errors, sentence structure and sentence fragments, etc. had been revised in revised manuscript. We would like to re-submit the revised manuscript for your consideration. We hope that the revision is acceptable for publication in your journal.

Point 12: References: Authors should consider revising the bibliography, as there are several incorrect citations. Indeed, according to the Journal’s guidelines, they should provide the abbreviated journal name in italics, the year of publication in bold, the volume number in italics for all the references.

 Response 12: Thank you for your careful review and valuable suggestions. We checked all the references and revised the bibliography which are incorrect citations. Besides, according to the Journal’s guidelines, we provide the abbreviated journal name in italics, the year of publication in bold, the volume number in italics for all the references. (Lines 484-704).

  • Li Y, Baptista R P, Kissinger J C. Noncoding RNAs in apicomplexan parasites: an update[J]. Trends in Parasitology, 2020, 36(10): 835-849.
  • Knopman D S, Amieva H, Petersen R C, et al. Alzheimer disease[J]. Nature reviews Disease primers, 2021, 7(1): 1-21.
  • Andersen J K. Oxidative stress in neurodegeneration: cause or consequence[J]. Nature medicine, 2004, 10(7): S18-S25.
  • Lima J A, Hamerski L. Alkaloids as potential multi-target drugs to treat Alzheimer's disease[J]. Studies in natural products chemistry, 2019, 61: 301-334.
  • Sharma C, Kim S, Nam Y, et al. Mitochondrial dysfunction as a driver of cognitive impairment in Alzheimer’s disease[J]. International Journal of Molecular Sciences, 2021, 22(9): 4850.
  • Von Bernhardi R, Eugenín J. Alzheimer's disease: redox dysregulation as a common denominator for diverse pathogenic mechanisms[J]. Antioxidants & redox signaling, 2012, 16(9): 974-1031.
  • Gitler A D, Dhillon P, Shorter J. Neurodegenerative disease: models, mechanisms, and a new hope[J]. Disease models & mechanisms, 2017, 10(5): 499-502.
  • Farkhondeh T, Forouzanfar F, Roshanravan B, et al. Curcumin effect on non-amyloidogenic pathway for preventing alzheimer’s disease[J]. Biointerface Research in Applied Chemistry, 2019, 9(4): 4085-4089.
  • Bălașa A F, Chircov C, Grumezescu A M. Body Fluid Biomarkers for Alzheimer’s Disease—An Up-To-Date Overview[J]. Biomedicines, 2020, 8(10): 421.
  • Khalil M, Teunissen C E, Otto M, et al. Neurofilaments as biomarkers in neurological disorders[J]. Nature Reviews Neurology, 2018, 14(10): 577-589.
  • Battaglia S, Garofalo S, di Pellegrino G. Context-dependent extinction of threat memories: influences of healthy aging[J]. Scientific reports, 2018, 8(1): 1-13.
  • Serafín V, Gamella M, Pedrero M, et al. Enlightening the advancements in electrochemical bioanalysis for the diagnosis of Alzheimer’s disease and other neurodegenerative disorders[J]. Journal of Pharmaceutical and Biomedical Analysis, 2020, 189: 113437.
  • Nazam F, Shaikh S, Nazam N, et al. Mechanistic insights into the pathogenesis of neurodegenerative diseases: towards the development of effective therapy[J]. Molecular and Cellular Biochemistry, 2021, 476(7): 2739-2752.
  • Cui Y, Yang M, Zhu J, et al. Developments in diagnostic applications of saliva in Human Organ Diseases[J]. Medicine in Novel Technology and Devices, 2022: 100115.
  • Brito-Aguilar R. Dementia around the world and the Latin America and Mexican scenarios[J]. Journal of Alzheimer's Disease, 2019, 71(1): 1-5.
  • Liang D, Lu H. Salivary biological biomarkers for Alzheimer’s disease[J]. Archives of oral biology, 2019, 105: 5-12.
  • Mollan S P, Davies B, Silver N C, et al. Idiopathic intracranial hypertension: consensus guidelines on management[J]. Journal of Neurology, Neurosurgery & Psychiatry, 2018, 89(10): 1088-1100.
  • Chávez-Gutiérrez L, Szaruga M. Mechanisms of neurodegeneration—Insights from familial Alzheimer’s disease[C]//Seminars in Cell & Developmental Biology. Academic Press, 2020, 105: 75-85.
  • Brazaca L C, Sampaio I, Zucolotto V, et al. Applications of biosensors in Alzheimer's disease diagnosis[J]. Talanta, 2020, 210: 120644.
  • Yao F, Zhang K, Zhang Y, et al. Identification of blood biomarkers for Alzheimer's disease through computational prediction and experimental validation[J]. Frontiers in neurology, 2019, 9: 1158.
  • Jack Jr C R, Bennett D A, Blennow K, et al. NIA‐AA research framework: toward a biological definition of Alzheimer's disease[J]. Alzheimer's & Dementia, 2018, 14(4): 535-562.
  • Tanaka M, Toldi J, Vécsei L. Exploring the etiological links behind neurodegenerative diseases: Inflammatory cytokines and bioactive kynurenines[J]. International Journal of Molecular Sciences, 2020, 21(7): 2431.
  • Lake J, Storm C S, Makarious M B, et al. Genetic and Transcriptomic Biomarkers in Neurodegenerative Diseases: Current Situation and the Road Ahead[J]. Cells, 2021, 10(5): 1030.
  • Kim K Y, Shin K Y, Chang K A. Brain-Derived Exosomal Proteins as Effective Biomarkers for Alzheimer’s Disease: A Systematic Review and Meta-Analysis[J]. Biomolecules, 2021, 11(7): 980.
  • Motataianu A, Barcutean L I, Maier S, et al. Cardiac Autonomic Neuropathy in Diabetes Mellitus Patients-Are We Aware of the Consequences[J]. Acta Medica Marisiensis, 2020, 66(1):1-7.
  • Leandrou S, Lamnisos D, Mamais I, et al. Assessment of Alzheimer’s disease based on texture analysis of the entorhinal cortex[J]. Frontiers in Aging Neuroscience, 2020, 12: 176.
  • Suppiah S, Didier M A, Vinjamuri S. The who, when, why, and how of PET amyloid imaging in management of Alzheimer’s disease—Review of literature and interesting images[J]. Diagnostics, 2019, 9(2): 65.
  • Monajjemi M. Molecular vibration of dopamine neurotransmitter: A relation between its normal modes and harmonic notes[J]. Biointerface Res. Appl. Chem, 2019, 9: 3956-3962.
  • Gupta J, Fatima M T, Islam Z, et al. Nanoparticle formulations in the diagnosis and therapy of Alzheimer's disease[J]. International journal of biological macromolecules, 2019, 130: 515-526.
  • Montgomery W, Goren A, Kahle-Wrobleski K, et al. Detection, diagnosis, and treatment of Alzheimer’s disease dementia stratified by severity as reported by caregivers in Japan[J]. Neuropsychiatric Disease and Treatment, 2018, 14: 1843.
  • Kim J, Campbell A S, de Ávila B E F, et al. Wearable biosensors for healthcare monitoring[J]. Nature biotechnology, 2019, 37(4): 389-406.
  • Farah R, Haraty H, Salame Z, et al. Salivary biomarkers for the diagnosis and monitoring of neurological diseases[J]. Biomedical journal, 2018, 41(2): 63-87.
  • Jasim H, Olausson P, Hedenberg-Magnusson B, et al. The proteomic profile of whole and glandular saliva in healthy pain-free subjects[J]. Scientific reports, 2016, 6(1): 1-10.
  • Jasim H, Carlsson A, Hedenberg-Magnusson B, et al. Saliva as a medium to detect and measure biomarkers related to pain[J]. Scientific reports, 2018, 8(1): 1-9.
  • Lee M, Guo J P, Kennedy K, et al. A method for diagnosing Alzheimer’s disease based on salivary amyloid-β protein 42 levels[J]. Journal of Alzheimer's Disease, 2017, 55(3): 1175-1182.
  • Ashton N J, Ide M, Schöll M, et al. No association of salivary total tau concentration with Alzheimer's disease[J]. Neurobiology of aging, 2018, 70: 125-127.
  • Kodintsev A N, Kovtun O P, Volkova L I. Saliva biomarkers in diagnostics of early stages of Alzheimer's disease[J]. Neurochemical Journal, 2020, 14(4):429-438.
  • 38[16] Li Y, Li R, Li X, et al. Effects of different aerobic exercise training on glycemia in patients with type 2 diabetes: A protocol for systematic review and meta-analysis [J]. Medicine, 2021, 100(18): e25615.
  • Almeida Pdel V, Gregio A M, Machado M A, et al. Saliva composition and functions: a comprehensive review[J]. The journal of contemporary dental practice,2008, 9:72–80.
  • Lee, Kwon, Shin, et al. Optimization of Saliva Collection and Immunochromatographic Detection of Salivary Pepsin for Point-of-Care Testing of Laryngopharyngeal Reflux[J]. Sensors, 2020, 20(1):325-333.
  • Kai D T, Kenny L, Frazer I H, et al. High‐risk human papillomavirus detection in oropharyngeal cancers: Comparison of saliva sampling methods[J]. Head & Neck, 2019, 41: 1484-1489.
  • Kara D, Bayrak N A, Volkan B, et al. Anxiety and salivary cortisol levels in children undergoing esophago-gastro-duodenoscopy under sedation[J]. Journal of pediatric gastroenterology and nutrition, 2019, 68(1): 3-6.
  • Cui Y, Zhang H, Zhu J, et al. Unstimulated Parotid Saliva Is a Better Method for Blood Glucose Prediction[J]. Applied Sciences, 2021, 11(23): 11367.
  • Cui Y, Zhang H, Zhu J, et al. Correlations of Salivary and Blood Glucose Levels among Six Saliva Collection Methods[J]. International Journal of Environmental Research and Public Health, 2022, 19(7): 4122.
  • Blennow K, Hampel H. CSF markers for incipient Alzheimer's disease[J]. The Lancet Neurology, 2003, 2(10): 605-613.
  • Buerger K, Ewers M, Pirttilä T, et al. CSF phosphorylated tau protein correlates with neocortical neurofibrillary pathology in Alzheimer's disease[J]. Brain, 2006, 129(11): 3035-3041.
  • Yousif T I, O’Reilly K, Nadeem M. Blood tests are not always helpful in predicting bacterial meningitis in children[J]. Sudanese journal of paediatrics, 2016, 16(2): 77.
  • Bellagambi F G, Lomonaco T, Salvo P, et al. Saliva sampling: Methods and devices. An overview[J]. TrAC Trends in Analytical Chemistry, 2020, 124: 115781.
  • Gupta S, Ahuja N. Salivary glands[J]. London, United Kingdom: IntechOpen, 2019: 63-76.
  • Pedersen A M L, Sørensen C E, Proctor G B, et al. Salivary secretion in health and disease[J]. Journal of oral rehabilitation, 2018, 45(9): 730-746.
  • Mohamed R, Campbell J L, Cooper-White J, et al. The impact of saliva collection and processing methods on CRP, IgE, and Myoglobin immunoassays[J]. Clinical and translational medicine, 2012, 1(1): 1-8.
  • Hernndez L M, Taylor M K. Salivary gland anatomy and physiology[J]. Salivary bioscience: foundations of interdisciplinary saliva research and applications. New York: Springer Nature, 2020: 11-20.
  • Punyadeera C. Human saliva as a tool to investigate intimate partner violence[J]. Brain, behavior, and immunity, 2012, 26(4): 541-542.
  • Niedrig M, Patel P, Abd El Wahed A, et al. Find the right sample: A study on the versatility of saliva and urine samples for the diagnosis of emerging viruses[J]. BMC infectious diseases, 2018, 18(1): 1-14.
  • Hartenbach F A R R, Velasquez É, Nogueira F C S, et al. Proteomic analysis of whole saliva in chronic periodontitis[J]. Journal of proteomics, 2020, 213: 103602.
  • Xin H, Katakowski M, Wang F, et al. MicroRNA-17–92 cluster in exosomes enhance neuroplasticity and functional recovery after stroke in rats[J]. Stroke, 2017, 48(3): 747-753.
  • Müller U C, Deller T, Korte M. Not just amyloid: physiological functions of the amyloid precursor protein family[J]. Nature Reviews Neuroscience, 2017, 18(5): 281-298.
  • Farah R, Haraty H, Salame Z, et al. Salivary biomarkers for the diagnosis and monitoring of neurological diseases[J]. Biomedical journal, 2018, 41(2): 63-87.
  • Ashton N J, Ide M, Schöll M, et al. No association of salivary total tau concentration with Alzheimer's disease[J]. Neurobiology of aging, 2018, 70: 125-127.
  • Sabbagh M N, Shi J, Lee M, et al. Salivary beta amyloid protein levels are detectable and differentiate patients with Alzheimer’s disease dementia from normal controls: Preliminary findings[J]. BMC neurology, 2018, 18(1): 1-4.
  • Blennow K, Hampel H, Weiner M, et al. Cerebrospinal fluid and plasma biomarkers in Alzheimer disease[J]. Nature Reviews Neurology, 2010, 6(3): 131-144.
  • Pekeles H, Qureshi H Y, Paudel H K, et al. Development and validation of a salivary tau biomarker in Alzheimer's disease[J]. Alzheimer's & Dementia: Diagnosis, Assessment & Disease Monitoring, 2019, 11: 53-60.
  • Reale M, Gonzales-Portillo I, Borlongan C V. Saliva, an easily accessible fluid as diagnostic tool and potent stem cell source for Alzheimer’s Disease: Present and future applications[J]. Brain Research, 2020, 1727: 146535.
  • Chandra A. Role of amyloid from a multiple sclerosis perspective: a literature review[J]. Neuroimmunomodulation, 2015, 22(6): 343-346.
  • Holmberg B, Johnels B, Blennow K, et al. Cerebrospinal fluid Aβ42 is reduced in multiple system atrophy but normal in Parkinson's disease and progressive supranuclear palsy[J]. Movement disorders: official journal of the Movement Disorder Society, 2003, 18(2): 186-190.
  • Lewczuk P, Lelental N, Spitzer P, et al. Amyloid-β 42/40 cerebrospinal fluid concentration ratio in the diagnostics of Alzheimer's disease: validation of two novel assays[J]. Journal of Alzheimer's disease, 2015, 43(1): 183-191.
  • Albahri A S, Alwan J K, Taha Z K, et al. IoT-based telemedicine for disease prevention and health promotion: State-of-the-Art[J]. Journal of Network and Computer Applications, 2021, 173: 102873.
  • Pekeles H, Qureshi H Y, Paudel H K, et al. Development and validation of a salivary tau biomarker in Alzheimer's disease[J]. Alzheimer's & Dementia: Diagnosis, Assessment & Disease Monitoring, 2019, 11: 53-60.
  • Park S A, Han S M, Kim C E. New fluid biomarkers tracking non-amyloid-β and non-tau pathology in Alzheimer’s disease[J]. Experimental & molecular medicine, 2020, 52(4): 556-568.
  • Ebneth A, Godemann R, Stamer K, et al. Overexpression of tau protein inhibits kinesin-dependent trafficking of vesicles, mitochondria, and endoplasmic reticulum: implications for Alzheimer's disease[J]. The Journal of cell biology, 1998, 143(3): 777-794.
  • Kent S A, Spires-Jones T L, Durrant C S. The physiological roles of tau and Aβ: implications for Alzheimer’s disease pathology and therapeutics[J]. Acta neuropathologica, 2020, 140(4): 417-447.
  • Iqbal K, Liu F, Gong C X. Tau and neurodegenerative disease: the story so far[J]. Nature reviews neurology, 2016, 12(1): 15-27.
  • Lau H C, Lee I K, Ko P W, et al. Non-invasive screening for Alzheimer’s disease by sensing salivary sugar using Drosophila cells expressing gustatory receptor (Gr5a) immobilized on an extended gate ion-sensitive field-effect transistor (EG-ISFET) biosensor[J]. PLoS One, 2015, 10(2): e0117810.
  • Shi M, Sui Y T, Peskind E R, et al. Salivary tau species are potential biomarkers of Alzheimer's disease[J]. Journal of Alzheimer's Disease, 2011, 27(2): 299-305.75.
  • Zetterberg H, Blennow K. Moving fluid biomarkers for Alzheimer’s disease from research tools to routine clinical diagnostics[J]. Molecular neurodegeneration, 2021, 16(1): 1-7.
  • Cano A, Turowski P, Ettcheto M, et al. Nanomedicine-based technologies and novel biomarkers for the diagnosis and treatment of Alzheimer’s disease: from current to future challenges[J]. Journal of Nanobiotechnology, 2021, 19(1): 1-30.
  • Lee J C, Kim S J, Hong S, et al. Diagnosis of Alzheimer’s disease utilizing amyloid and tau as fluid biomarkers[J]. Experimental & molecular medicine, 2019, 51(5): 1-10.
  • Pawlowski M, Meuth S G, Duning T. Cerebrospinal fluid biomarkers in Alzheimer’s disease—From brain starch to bench and bedside[J]. Diagnostics, 2017, 7(3): 42.
  • Niemantsverdriet E, Valckx S, Bjerke M, et al. Alzheimer’s disease CSF biomarkers: Clinical indications and rational use[J]. Acta neurologica belgica, 2017, 117(3): 591-602.
  • Vogel J W, Iturria-Medina Y, Strandberg O T, et al. Spread of pathological tau proteins through communicating neurons in human Alzheimer’s disease[J]. Nature communications, 2020, 11(1): 1-15.
  • Jin M, Cao L, Dai Y. Role of neurofilament light chain as a potential biomarker for Alzheimer's disease: a correlative meta-analysis[J]. Frontiers in Aging Neuroscience, 2019, 11: 254.
  • Battaglia S, Harrison B J, Fullana M A. Does the human ventromedial prefrontal cortex support fear learning, fear extinction or both? A commentary on subregional contributions[J]. Molecular Psychiatry, 2021: 1-3.
  • Tanaka M, Vécsei L. Editorial of Special Issue “Crosstalk between depression, anxiety, and dementia: comorbidity in behavioral neurology and neuropsychiatry [J]. Biomedicines, 2021, 9(5): 517.

Point 13: Overall, the manuscript contains 5 figures, 2 tables and 39 references. In my opinion, the number of references it is dramatically low for an original research article, and this prevents the possibility of publishing it in this form. References should be more than 50 for original research articles. However, the manuscript might carry important value describing how saliva biomarkers can be quantified and used to diagnose AD.

Response 13: Thank you for your careful review and valuable suggestions. In order to make this study clearer to the reader, and to provide a better and more accurate background to this study, we added more evidence to back our claims, especially in the introduction of the review. Now, the literature was added to 83 articles for a literature review, as followed. (Lines 484-704).

Point 14: I hope that, after these careful revisions, the manuscript can meet the Journal’s high standards for publication. I am available for a new round of revision of this article.

Best regards,

Reviewer

Response 14: Thank you for your careful review and valuable suggestions. We would like to express our sincere thanks again to you for the constructive and positive comments on our manuscript entitled “Validation of whole and glandular saliva as a biomarker for Alzheimer’s disease diagnosis” (Manuscript ID: brainsci-1676756). We have studied the comments of both the editor and reviewers carefully and tried our best to revise the paper accordingly. Point-to-point replies are included in this coverletter. Besides, we have optimized the article again this time, the grammar errors, sentence structure and sentence fragments, etc. had been revised in revised manuscript. We would like to re-submit the revised manuscript for your consideration. We hope that the revision is acceptable for publication in your journal.

Round 2

Reviewer 3 Report

I am very pleased to see that the authors have welcomed many suggestions and have clarified several of the questions I raised in my first round of this review. I believe that this original research article does an excellent work describing how saliva biomarkers can be quantified and used to diagnose Alzheimer’s disease.

I only have one more minor comment, since there is a discrepancy between the references in your manuscript and those actually reported in the reference list. Indeed, in the revised manuscript, when mentioning how the role of the ventromedial prefrontal cortex in the processing of safety threat information should also be included as a relevant risk factor in the diagnosis of AD, the authors stated that the manuscript was edited as suggested. However, one of the two indicated studies (reference n. 83) was wrongly cited: please, correct it with the proper citation (https://doi.org/10.17219/acem/146756) and update the Reference list accordingly to changes in the main text of the revised manuscript.

Overall, this is a timely and needed paper, and I believe that, after this final issue, the manuscript meets the Journal’s high standards for publication. 

I am always available for other reviews of such interesting and important articles.

Thank You for your work.

Author Response

Point 1: I am very pleased to see that the authors have welcomed many suggestions and have clarified several of the questions I raised in my first round of this review. I believe that this original research article does an excellent work describing how saliva biomarkers can be quantified and used to diagnose Alzheimer’s disease.

Response 1: Thank you for your careful review and valuable suggestions. We would like to express our sincere thanks again to you for the constructive and positive comments on our manuscript entitled “Validation of whole and glandular saliva as a biomarker for Alzheimer’s disease diagnosis” (Manuscript ID: brainsci-1676756). We have optimized the article again this time, the grammar errors, sentence structure and sentence fragments, etc. had been revised in revised manuscript. Besides, all the references have been checked and revised

Point 2: I only have one more minor comment, since there is a discrepancy between the references in your manuscript and those actually reported in the reference list. Indeed, in the revised manuscript, when mentioning how the role of the ventromedial prefrontal cortex in the processing of safety threat information should also be included as a relevant risk factor in the diagnosis of AD, the authors stated that the manuscript was edited as suggested. However, one of the two indicated studies (reference n. 83) was wrongly cited: please, correct it with the proper citation (https://doi.org/10.17219/acem/146756) and update the Reference list accordingly to changes in the main text of the revised manuscript. Overall, this is a timely and needed paper, and I believe that, after this final issue, the manuscript meets the Journal’s high standards for publication. I am always available for other reviews of such interesting and important articles. Thank You for your work.

Response 2: Thank you for your careful review and valuable suggestions. As the reviewer said, there was a discrepancy between the references in the manuscript and those actually reported in the reference list. So, all the references have been checked and revised. Besides, in the revised manuscript, "when mentioning how the role of the ventromedial prefrontal cortex in the processing of safety threat information should also be included as a relevant risk factor in the diagnosis of AD", this part was corrected to the proper citation with (https://doi.org/10.17219/acem/146756) and  the Reference list was updated accordingly to changes in the main text of the revised manuscript. (Lines 449-457).

However, our study has limitations. First of all, when studying several biomarkers, they have not been compared with gold standards such as CSF concentration and imaging. Although this study was conducted in patients who have already been diagnosed, the saliva biomarkers that reflect the current situation cannot be compared with image data obtained in some cases a few years ago. Therefore, the patient’s saliva biomarkers should be diagnosed when imaging or CSF is obtained for AD comparison to evaluate their salivary biomarkers. In addition, clinical studies have described the excellent diagnostic performance of saliva biomarkers. Although this study provides the best performance for early diagnosis of patients with cognitive problems and suspected AD, due to the small amount of data, further development is still needed, including validity testing, retesting, and multifactor testing. Finally, there are still many problems in this study that need to be resolved and further explored. For example, the submandibular glands and sublingual glands are closely located, so it is difficult to separate saliva from these glands with certainty, which was why saliva is collected from both glands in the current study. How to distinguish sublingual saliva from submandibular saliva is also a direction that needs further research. Finally, various risk factors (e. g. the role of ventromedial prefrontal cortex in the processing of safety-threat information and their relative value [82]) should also be included in the diagnosis of AD in saliva to make the diagnosis more accurate. Besides, providing a deeper understanding of human learning neural networks, particularly on human processing of safety crucial role is also the focus of future research. It might also contribute to the advancement of alternative, more precise and individualized treatments for psychiatric disorders [83].

This manuscript is a resubmission of an earlier submission. The following is a list of the peer review reports and author responses from that submission.